# Impact of the Traditional Lecture Teaching Method and Dalcroze’s Body Rhythmic Teaching Method on the Teaching of Emotion in Music—A Cognitive Neuroscience Approach

**DOI:** 10.3390/brainsci15121253

**Published:** 2025-11-21

**Authors:** Qiong Ge, Xu Li, Huiling Zhou, Meiqi Yu, Jie Lin, Quanwei Shen, Jiamei Lu

**Affiliations:** 1School of Psychology, Shanghai Normal University, Shanghai 200234, China; 1000497860@smail.shnu.edu.cn (Q.G.); lxuthus@shnu.edu.cn (X.L.); 1000550026@smail.shnu.edu.cn (H.Z.); 2College of Modern Science and Technology, China Jiliang University, Hangzhou 321000, China; xdkjxy@cjlu.edu.cn; 3School of Music, Zhejiang Normal University, Jinhua 321004, China; livia888899@zjnu.cn; 4Department of Psychology, Hubei University of Medicine, Shiyan 442000, China; psywe@hbmu.edu.cn

**Keywords:** music emotion processing, lecture teaching, body rhythm teaching, fNIRS hyperscanning

## Abstract

**Background:** Although the Shared Affective Movement Experience (SAME) model suggests the crucial role of imitation and synchronization in music-induced emotion, their application in teaching settings remains largely unexplored. **Objectives:** This study compared the “Body Rhythm Teaching Method,” based on the principle of mimicking musical elements through bodily movements, with traditional lecture-based instruction. It examined the effects of both teaching approaches on brain activation patterns, measured via functional Near-Infrared Spectroscopy (fNIRS) hyperscanning and instructional outcomes (assessed through musical emotion processing and teaching quality evaluations). The aim was to investigate their efficacy in enhancing students’ musical emotional processing abilities. **Methods:** A total of 3 teachers and 103 student participants were randomly assigned to the lecture teaching group (*n* = 35), the body rhythm teaching group (*n* = 35), or the control group (*n* = 33). The musical materials used across all three groups were identical, with only the teaching methods differing. fNIRS hyperscanning imaging was employed throughout the process to record brain activity. **Results:** Results indicate that the body rhythm group significantly outperformed other groups in both behavioral and neural metrics. Specifically, during the post-test music-listening phase, participants in this group not only reported higher emotional arousal but also exhibited stronger activation levels in the bilateral frontopolar cortex (FPC) associated with multisensory integration—both significantly higher than those in the lecture group and control group. Furthermore, during instruction, students in the body rhythm group rated teaching quality higher and exhibited significantly stronger teacher–student IBS across multiple brain regions involved in socio-emotional processing. These included the left orbitofrontal cortex (lOFC) for interoceptive emotion processing, the left frontopolar cortex (lFPC) for multisensory integration, and the right superior temporal gyrus (rSTG) for social interaction. In contrast, the lecture teaching group only showed significantly higher emotional valence ratings compared to the control group. **Conclusions:** This study confirms the role of imitation and synchronization mechanisms in the SAME model for music-induced emotional responses, providing a neuroscientific basis for teaching practice.

## 1. Introduction

### 1.1. Objective and Rationale

Consider a scenario in which, after listening to a piece of music in a music class, some students evoke a rich emotional response and dance to the music, while others do not respond or even recognize the emotion that the music is trying to convey. This phenomenon has forced teachers to think about how to help students improve their emotional processing ability when listening to music, which has been a great concern in the field of music education psychology [1].

Emotional processing of music consists of the experience of music emotions (sensation) and the recognition of music emotions (perception) [2]. Accordingly, the objectives of music education manifest in two specific objectives: first, cultivating the ability to accurately identify emotions in music (such as distinguishing joy from sorrow [3]), and second, fostering genuine emotional experiences to prevent emotional disengagement [4]. These dimensions can be quantitatively assessed using the well-established valence–arousal model [5]. Research confirms that music-evoked emotions hold significant value, contributing not only to short-term mood regulation but also to long-term emotional development and social competence [2,6].

Addressing the lack of emotional response requires moving beyond attributions to fixed factors like musical features or listener traits. Instead, we focus on the process through which music induces emotions and how this process can be pedagogically influenced—a perspective offering greater potential for intervention [7]. We investigate two contrasting teaching methods aimed at modifying this emotional induction process. The traditional lecture teaching method directly transmits knowledge about musical emotions through verbal explanation, training students to identify perceivable emotional cues such as tempo, melody, and harmony [8]. In contrast, the body rhythm teaching method (based on Dalcroze Eurhythmics [9]) uses synchronized movement to foster an embodied understanding of music, enabling learners to physically experience musical elements and their emotional correlates [10].

In the following chapters, we will examine and compare two pedagogical approaches to improve the phenomenon of students not feeling emotions in music teaching: by directly explaining to students the knowledge of emotions in music (lecture teaching group) and by mimicking musical elements through body movements (body rhythm teaching group).

### 1.2. Theoretical Frameworks for Teaching Music Emotion

The lecture teaching method aligns with the Meaningful Reception Learning Theory [11], which posits that the process of teaching knowledge to students should emphasize students’ subjective initiative (i.e., students clarify the content of their learning), and reach the process of grasping socio-historical experience in the form of individual experience (i.e., focusing on the students’ original state of knowledge). When using it in the teaching of musical emotions, teachers follow a “teach-receive” model, verbally explaining to students the cultural messages and emotional elements of a musical work, and encouraging learners to make connections between old and new knowledge in the context of their own experiences [8]. Empirical studies support its efficacy in delivering systematic knowledge in arts education [12,13,14] and in parsing complex musical structures [15].

Conversely, the body rhythm method is grounded in the Shared Affective Movement Experience (SAME) model [16], which posits that movement imitation and synchronization are core mechanisms for musical empathy and emotional resonance. While traditional Dalcroze pedagogy emphasizes rhythmic internalization [17], our SAME-based practice extends to interpreting broader musical characteristics—such as melody, pitch, timbre, and dynamics—through synchronized movement. This establishes dynamic connections between movement and musical elements, thereby stimulating emotional resonance. Experimental evidence has demonstrated the benefits of the Dalcroze teaching method for enhancing students’ rhythmic perception [18], as well as improved motor coordination and rehabilitation outcomes [19]. However, despite these established motor and perceptual benefits, practical music instruction often prioritizes technical skill acquisition over emotional experience [20], leaving the method’s specific efficacy in emotional teaching—particularly in comparison to lecture teaching instruction—underexplored at a neurocognitive level.

### 1.3. Comparison of Lecture Teaching Method and Body Rhythm Teaching Method on the Process of Music-Evoked Emotions in Students

The lecture method directly imparts knowledge of musical emotions through verbal instruction, whereas the body rhythm teaching method guides embodied perception of emotions via kinesthetic engagement. Though both enhance emotional processing, they operate through distinct music–emotion induction mechanisms [21]. Lecture teaching relies on unidirectional semantic decoding, forming abstract conceptual representations (disembodied cognition); the body rhythm teaching method transforms musical elements into embodied cognition through movement imitation, surpassing passive observation to dynamically interpret musical structures via improvisational expression. This study thus validates both methods’ efficacy in music–emotion teaching, probing their divergent emotion-evoking mechanisms.

In addition, although both lecture teaching and body rhythm teaching methods may have triggered cognitive processing during students’ listening to music to generate emotions, these two teaching methods resulted in the three factors of teaching (teacher, students, and teaching materials [22]) showing different relational orientations in the teaching and learning process: the teacher-centered lecture method prioritizes knowledge transmission, whereas the student-centered body rhythm method facilitates active, embodied discovery. Consequently, music is treated as an external object in the former, but is experienced as an embodied extension of the self in the latter.

Given the scarcity of empirical research on these emotional teaching methods and oversimplified assessment indicators in existing studies, a systematic examination of their effects is required, focusing on music-induced emotion outcomes and interactions among core elements (teachers, students, and materials). Functional Near-Infrared Spectroscopy (fNIRS) hyperscanning is an effective tool for measuring emotional activation, given its validated sensitivity to auditory and higher-order cortical processing [23] and its tolerance to movement artifacts [24]. This technology enables the investigation of neural response synchronization during teacher–student interactions [25]. The Interpersonal Brain Synchronization (IBS) serves as a robust neural metric for quantifying the dynamic interplay between teachers, students, and instructional materials. Given that IBS predicts student engagement [26], academic performance [27], and learning transfer [28], fNIRS hyperscanning is well-suited to capture the neural correlates of teaching effectiveness and interaction in naturalistic settings.

### 1.4. The Neural Basis for Lecture Teaching and Body Rhythm Teaching

We employ cognitive neuroscience to measure brain activity during emotional teaching, hypothesizing that distinct cognitive processes occur across different methods. Our primary question is: what neural advantages does the embodied cognition of body rhythm teaching hold over the disembodied cognition of lecture teaching, and where are these differences localized? Existing research indicates that several subregions of the prefrontal cortex (PFC) play a key role in emotion-related instructional processing [23]. Specifically, the frontopolar cortex (FPC; BA10) is involved in the cognitive regulation of emotions [29], responsible for integrating attention allocation to external stimuli and maintaining internally generated cognitive processes [30]; the dorsolateral prefrontal cortex (dlPFC; BA9/46) is crucial for higher-order executive functions such as logical reasoning and imagination in emotional contexts [31]; while the orbitofrontal cortex (OFC; BA11) is closely associated with the processing of interoceptive emotional signals and subjective emotional experiences [32].

In emotional teaching, the right temporo-parietal junction (rTPJ) is another crucial region [27]. As a hub for social cognition, it underpins Theory of Mind and empathy by inferring mental states and facilitating emotional sharing [33]. A key subsystem, the superior temporal gyrus (STG), is vital for social auditory-motor integration due to its mirror neuron properties. The STG processes auditory emotional signals and integrates them with visual cues to decode social intent [34]. Collectively, the rTPJ and STG form a socio-emotional network critical for distinguishing the neural effects of different teaching methods.

### 1.5. The Current Study

In this study, university students were asked to participate in a lesson on the appreciation of traditional Chinese Pipa music (see Appendix A for more Pipa music). Choosing only Pipa music has several advantages, such as avoiding potential confusion from other instrumental timbres and lyrics. Additionally, we controlled for preference and familiarity with Pipa music. In addition, having the subjects’ cultural background match that of the selected music helped control for two potential confounding variables, music genre and cultural factors, thereby reducing interference with the study’s results. Subjects enjoyed Pipa music through the traditional lecture teaching method used by the instructor (lecture teaching group), the Dalcroze body rhythm teaching method (body rhythm teaching group), and the method used by the no-teaching group (control group).

The primary objective of this study was to determine whether different teaching methods were effective in enhancing students’ musical emotional processing and to identify which one was rated higher in terms of teaching quality. We expected to find the effectiveness and differences among the three groups in subjective ratings of musical emotional processing, and to explore whether there were differences between the two teaching methods in terms of teaching evaluation. For the second objective, we aimed to investigate whether there were differences in brain activity among the three groups and to determine the significance of any observed differences.

**Hypothesis** **1.**
*Teaching Effectiveness and Teaching Quality Evaluations.*


**Hypothesis** **1a.**
*We predicted that both the lecture teaching group and the body rhythm group would outperform the control group in musical emotional processing, with the body rhythm group performing best.*


**Hypothesis** **1b.**
*We hypothesized that the body rhythm teaching group would rate the quality of instruction higher than the lecture teaching group in the teaching evaluation.*


**Hypothesis** **2.**
*Brain Activity.*


**Hypothesis** **2a.**
*We expected that during the post-test phase of instruction, the lecture teaching group and the body rhythm teaching group would show greater brain activity than the control group for specific domains related to emotional processing, with the body rhythm group showing the most activation of brain activity.*


**Hypothesis** **2b.**
*We expected that the lecture teaching group and the body rhythm teaching group would exhibit greater teacher–student inter-brain synchronization than the control group in specific domains related to emotional processing and social interaction during the teaching phase, with the body rhythm group showing the highest teacher–student IBS activation.*


## 2. Method

### 2.1. Participants and Design

A G*Power 3.1.9 analysis (1 − β = 0.95, α = 0.05, f = 0.25) indicated 57 participants were required. We recruited 106 Chinese university students. A total of 3 were excluded due to device contact issues, leaving 103 (65 females, age 22.61 ± 2.39) to complete the experiment. Teachers and learners were unacquainted with each other before participating in the study. Learners were randomly split into three groups: the lecture teaching (35 learners, 20 females, age 22.54 ± 2.47), the body rhythm teaching (35 learners, 20 females, age 23.03 ± 2.33), and the control group (33 learners, 24 females, age 22.24 ± 2.37). Three female music teachers (age 27.33 ± 1.15, with over 5 years of teaching experience) participated. All subjects possessed normal hearing and vision (or achieved normal levels with correction), had no psychiatric disorders, and had received no formal music training beyond school music courses (exceeding one year). All participants signed informed consent forms and received compensation upon completion of the experiment.

### 2.2. Materials

Materials included demographic questionnaires, music emotional processing scales, teaching quality evaluation scales, and music materials and ratings. All materials were in Mandarin.

#### 2.2.1. Demographic Questionnaire

A demographic questionnaire was used to collect information about the participants’ gender, grade level, age, major, level of preference for Pipa music, and daily exposure to Pipa music.

#### 2.2.2. Musical Emotional Processing Scale, MPS

The assessment of music emotion recognition and music emotion experience was measured using a self-report scale based on the two-dimensional model of valence and arousal [35]. These two dimensions were guided and distinguished through different instructions and questioning methods [36]. Previous research indicates that sequence differences do not yield significant variations in outcomes [37]. Therefore, the following four questions are presented in a fixed order.

Music emotion recognition and experience are measured through two dimensions: valence and arousal, each assessed by two items. Emotion recognition items evaluate the valence and intensity of emotions conveyed by the music, while emotion experience items assess the actual valence and intensity of emotions felt by listeners. All items are rated on a 7-point scale (1 = negative/calm, 7 = positive/excited). The scale demonstrated good reliability, with Cronbach’s α increasing from 0.78 (pre-test) to 0.83 (post-test).

#### 2.2.3. Teaching Quality Evaluation Scale, TS

The Teaching Quality Evaluation Scale, adapted from Cai et al. (2014) [38], was designed to measure students’ assessment of teaching quality at the end of each lesson. Some items were modified or deleted because they did not apply to the context of this experiment (e.g., “Whether the teacher was strictly managed and the class was in good order,” and “Whether the teacher uses board notes and modern teaching techniques”). The final scale consisted of six items on a 5-point Likert scale (1 = “poor” to 5 = “Excellent”). Scores from these six items were summed (range: 6–30). Cronbach’s α was 0.91.

#### 2.2.4. Music Materials and Ratings

Materials Selection and Preparation: Ten pure Pipa instrumental pieces were sourced from major music platforms (e.g., Kugou Music, QQ Music), all composed in the pentatonic mode and performed by renowned artists. Following evidence that extended musical excerpts are required to induce emotions [39], 120 s segments were extracted from each piece (average loudness: 65–75 dB; sampling frequency: 44.1 kHz; bit depth: 16 bits). Music excerpts were categorized by valence and energy arousal [40] to ensure emotional validity. Both positive and negative music were included due to their distinct educational roles: positive music fosters pleasure and engagement [41], while negative music supports catharsis and creativity [42,43].

Rating of musical materials: Thirty-two non-music majors rated the emotional valence (1 = negative to 7 = positive) and arousal (1 = calm to 7 = excited) of musical excerpts on a 7-point scale, excluding works with above-moderate familiarity. Two positive- and two negative-valence pieces were selected (Appendix A), each segmented into four 30 s excerpts for analysis and instruction phases.

### 2.3. fNIRS Regions of Interest

The fNIRS device and channel layout are presented in Appendix A and Figure 1. Given that the prefrontal cortex is closely associated with emotional processing [44], the right temporo-parietal joint area is an important region for social interaction learning [27]. Therefore, we selected the PFC and right TPJ regions as regions of interest (ROI; see Table 1).

### 2.4. Procedure

Experiments took place in a quiet, soundproof lab with stable lighting. E—Prime 3.0 on a 19-inch computer compiled the experimental program and recorded responses automatically, and an external splitter linked to two headphones played music. Students and teachers were randomly paired and assigned to groups (Figure 2a). After obtaining participants’ informed consent, they completed the sequential stages as shown in Figure 2b.

During the preparation phase, both the instructor and student participants were seated in front of a computer and instructed to assume a comfortable, stable posture. The experimenter then assisted them in wearing the fNIRS headcap and headphones, with volume levels individually calibrated for clarity. Optical sources and detectors were positioned according to the predefined channel layout template. The optode–scalp contact was carefully optimized until stable and high-quality signals were confirmed across all channels. Following equipment setup, a 3 min resting-state baseline was recorded, during which participants were instructed to remain still, relax, and clear their minds.

To emulate the natural flow of classroom instruction, the experiment was divided into three distinct phases that reflect the common structure of a music appreciation course.

During the holistic listening phase, participants listen to the complete piece of music for the first time. Throughout the listening process, their eyes must remain focused on the fixation point displayed on the screen. Upon completion of the music, they subjectively evaluate the valence and arousal level of their emotional processing (emotion recognition and emotional experience) for that segment. Following a 20 s rest period, they proceed to the next phase.

During the analytical teaching phase, the teacher selects one of two instructional methods. The specific teaching design includes instructional prompts and content, lasting approximately 4 min (see Figure 2c). Partial learning (breaking music into segments for sequential study) generates more interaction and yields better learning outcomes than holistic learning (studying a piece of music in its entirety) [45]. Therefore, teachers decomposed each piece into four segments for instruction, alternating between explanation and listening. Specific content was determined pre-class through discussions with the teaching team based on the chosen method’s characteristics (designs linked musical features to body movements, specifically involving hands, arms, and other upper-body actions; see Appendix A). Throughout instruction, teachers maintained neutral facial expressions, consistent volume, and calm intonation. After each session, student participants completed evaluations of teaching quality. Following a 20 s break, the next phase commenced.

During the comprehensive listening phase, student participants applied the knowledge or methods acquired during the analytical instruction phase to revisit the musical piece. Following the conclusion of the music, they completed subjective ratings of the valence and arousal levels associated with emotional processing (emotion recognition and emotional experience) for this segment of the music. The whole time, in order to avoid the influence of the teacher’s presence on the students’ keystrokes. The teacher’s attention was focused on the lesson plan and the explanation of the teaching methodology and was not involved in any of the students’ keystroke responses.

The entire experiment lasted about 40 min, with a total of 4 (positive 2, negative 2) pieces of music, each of which averaged 120 s in length, and the music sequence was counterbalanced between subjects. All student participants received a compensation of 40 RMB upon completion of the entire experiment. The three instructors involved in teaching were also provided with a corresponding fixed honorarium. All details regarding compensation were clearly communicated to every participant as part of the informed consent procedure before the commencement of the study. The experimental procedures were reviewed by the Ethics Committee of Shanghai Normal University.

### 2.5. Data Analysis

#### 2.5.1. Subjective Evaluation Data Analysis

Subjective rating data were analyzed using SPSS 26.0 (largely consistent with normality and chi-square, see Appendix A). First, a three-way repeated-measures analysis of variance (ANOVA) with 3 (teaching methods: lecture, body rhythm, control) × 2 (music validity: positive, negative) × 2 (time: pre, post) was conducted on the validity and arousal ratings of music emotion recognition and experience to examine the differences in teaching methods on the pre- and post-tests of music emotion processing. Based on the two-dimensional theory of emotional processing (valence and arousal), we divided the four behavioral dependent variables of musical emotion processing into two conceptually independent “statistical families” and performed false discovery rate (FDR) correction independently within each family. Valence Family: This family includes all statistical tests related to the assessment of emotional pleasantness. Specifically, it includes all *p*-values for key effects conducted on the two dependent variables: “emotion recognition valence” and “emotion experience valence”. Arousal Family: This family encompasses all statistical tests related to the assessment of emotional arousal intensity. Specifically, it includes all *p*-values for key effects conducted on the two dependent variables: “emotion recognition arousal” and “emotion experience arousal”. Second, a two-way repeated-measures ANOVA with 2 (teaching method: lecture, body rhythm) × 2 (music valence: positive, negative) was conducted to examine the differences in teaching quality scores across teaching methods.

#### 2.5.2. fNIRS Data Analysis

Preprocessing: First, raw data were imported into nirsLABv201904 (analysis software based on the NIRScout system). Signal quality of raw channel data was assessed using coefficients of variation (CVs), and channels with CV exceeding 15% were excluded [46,47]. Subsequently, data preprocessing was performed using Matlab R2013b functions and the Homer2 toolkit [48]: (1) Convert raw optical density (OD) data to oxyhemoglobin (HbO) and deoxyhemoglobin (HbR) concentrations; (2) remove global motion artifacts using the hmrMotionCorrectPCA function (nSV = 0.80) [49]; (3) correct motion artifacts in optical density signals using the Correlation-Based Signal Improvement (CBSI) method (turnon = 1); (4) the light intensity signal was filtered using a 0.01–0.08 Hz bandpass filter and a 0.15–0.3 Hz band-reject filter (to remove respiratory noise) [24]; (5) based on the modified Beer–Lambert law, convert optical density signals into changes in oxyhemoglobin and deoxyhemoglobin concentrations [50]. Previous studies have demonstrated that oxyhemoglobin exhibits greater sensitivity and is more widely applied in fNIRS superscanning studies [51]. Therefore, this experiment primarily focuses on changes in oxyhemoglobin concentration.

Brain Activation: The prefrontal cortex and right temporo-parietal junction were further subdivided into 14 regions of interest (ROIs; see Section 2.3). The mean blood oxygenation level-dependent response signal across all channels within each ROI was calculated to derive its composite activation level, serving as the basic unit for subsequent statistical analysis. Subsequently, a three-factor repeated-measures ANOVA was conducted on the beta values across all ROIs for the three groups: 3 (teaching method: lecture teaching group, body rhythm group, control group) × 2 (music valence: positive, negative) × 2 (time: pre-test, post-test), and corrected for multiple comparisons using the false discovery rate (FDR) method (significance level set at *p* < 0.05).

Wavelet transform coherence analysis: The computation of inter-brain synchrony focused on specific paired correspondences between teachers and students across 14 ROIs. We employed the wavelet transform coherence algorithm [52] to calculate the neural activity synchrony for each paired homologous teacher–student brain region. This technique effectively characterizes the coupling strength of dual-brain neural signals across the time–frequency domain and has been well-established in hyperscanning research paradigms [52]. In our specific analysis, we first computed the frequency-domain coherence for all channel pairs during both the resting phase and the teaching task phase. Following temporal averaging, we obtained the WTC matrices for each phase. To isolate task-induced synchrony changes from inherent inter-individual activity similarities, we defined the teaching-phase brain-to-brain synchrony metric as the difference between task-related coherence and baseline coherence, using resting-state brain synchrony (derived from stable data collected 120 s prior to music listening) as the baseline [53]. To identify effective analysis bands, this study performed systematic frequency-domain screening. An initial scan across the entire 0.027–1 Hz band revealed extensive signal distribution. 

Building on this, we applied a band-pass filter to suppress physiological noise—such as heartbeat (0.8–2.5 Hz [54]) and respiratory noise (0.20–0.30 Hz [55])—while preserving the frequency band of interest. This approach incorporated prior evidence indicating that enhanced inter-brain synchronization during teacher–student interactions predominantly occurs above 0.025 Hz [52]. Building on this, we applied a band-pass filter to maximize suppression of physiological noise, such as heartbeat (0.8–2.5 Hz [54]) and respiratory noise (0.20–0.30 Hz [55]),while preserving the frequency band of interest. This approach incorporated prior evidence indicating that enhanced inter-brain synchronization during teacher–student interactions predominantly occurs above 0.025 Hz [52]. Consequently, the core analysis band was precisely defined as 0.036–0.2 Hz (corresponding to a period of 5–28 s). Additionally, all inter-brain synchronization values underwent Fisher-z transformation prior to analysis. Subsequently, a 3 (teaching method: lecture group, body rhythm group, control group) × 2 (musical valence: positive, negative) two-factor repeated-measures ANOVA was conducted on the mean IBS values across the three groups. Multiple comparisons were corrected using the false discovery rate (FDR) method (significance level set at *p* < 0.05).

## 3. Results

An analysis of variance was conducted on the demographic variables and pre-instruction music emotion experience ratings across the three groups. The results indicated no significant differences among the groups on these measures (*F*(2, 100) ≤ 1.18, *p* ≥ 0.256, *η_p_*^2^ ≤ 0.02; see Appendix A). These findings confirm that the random assignment of participants to the three groups was successful.

### 3.1. Hypothesis 1: Do the Groups Differ in Their Evaluation of Teaching Effectiveness and Teaching Quality?

Hypothesis 1a proposed that both the lecture and body rhythm groups would outperform the control group in the post-test (Listening Stage 2; Figure 2b), with the body rhythm group expected to show the highest performance in musical emotional processing. Table 2 and Figure 3 present the means, standard deviations, and visual comparisons across the three groups. A three-way repeated-measures ANOVA was conducted to analyze group differences in musical emotional processing scores.

For emotion recognition valence, a significant three-way interaction (*F*(2, 100) = 19.27, *p* < 0.001, *η_p_*^2^ = 0.28), simple effects analysis using Bonferroni was performed. For positive music (post-test), consistent with H1a, both lecture (5.83 ± 0.16; *p* = 0.001) and body rhythm (5.86 ± 0.16; *p* = 0.001) scored higher than control (5 ± 0.16), but lecture vs. body rhythm was non-significant (*p* = 1), inconsistent with H1a. For negative music (post-test), body rhythm (2.31 ± 0.15) was significantly lower than control (*p* < 0.001), consistent with H1a, but lecture (2.76 ± 0.15) vs. control (3.29 ± 0.16; *p* = 0.053) and lecture vs. body rhythm (*p* = 0.13) were non-significant, inconsistent with H1a. For arousal, a significant time × method interaction (*F*(2, 100) = 6.42, *p* = 0.002, *η_p_*^2^ = 0.11) was shown at post-test: body rhythm (5.17 ± 0.11) was significantly higher than control (4.74 ± 0.12; *p* = 0.03), consistent with H1a, but lecture (5.01 ± 0.11) vs. control (*p* = 0.29) and lecture vs. body rhythm (*p* = 0.99) were non-significant, inconsistent with H1a.

For emotion experience valence, a significant three-way interaction (*F*(2, 100) = 9.67, *p* < 0.001, *η_p_*^2^ = 0.16) prompted simple effects tests. For positive music (post-test), consistent with H1a, lecture (5.47 ± 0.15; *p* = 0.02) and body rhythm (5.84 ± 0.15; *p* < 0.001) scored higher than control (4.91 ± 0.15), but lecture vs. body rhythm was non-significant (*p* = 0.22), inconsistent with H1a. For negative music (post-test), inconsistent with H1a, lecture (3.19 ± 0.16) vs. control (3.21 ± 0.17; *p* = 1), body rhythm (2.78 ± 0.16) vs. control (*p* = 0.21), and lecture vs. body rhythm (*p* = 0.25) were all non-significant. For emotion experience arousal, a significant time × method interaction (*F*(2, 100) = 8.57, *p* < 0.001, *η_p_*^2^ = 0.15) was shown at post-test: inconsistent with H1a, lecture (5.19 ± 0.13) vs. control (4.89 ± 0.13; *p* = 0.34) was non-significant; consistent with H1a, body rhythm (5.64 ± 0.13) was significantly higher than control (*p* < 0.001) and body movement was significantly higher than lecture (*p* = 0.04). See Appendix A for other results.

Hypothesis 1a, which proposed that both teaching methods would outperform the control group, was partially supported. While both groups showed superior performance in valence-related indicators (emotion recognition and experience), only the body movement group demonstrated significantly higher arousal scores—in both recognition and experience—compared to the control group. No significant difference in arousal was found between the lecture group and the control condition. These results reinforce prior evidence that emotion-focused instruction enhances musical emotion processing more effectively than passive listening. Furthermore, the body movement group exhibited significantly greater emotional experience arousal than the lecture group, consistent with embodied cognition theories, which suggests that physical engagement facilitates deeper emotional involvement than purely verbal instruction.

Hypothesis 1b was that the body rhythm teaching group would score higher than the lecture teaching group on the teaching quality evaluation. A repeated-measures ANOVA confirmed a significant main effect of teaching method, *F*(1, 68) = 11.97, *p* = 0.001, *η_p_*^2^ = 0.15, with the body rhythm group rated significantly higher than the lecture group, consistent with the hypothesis. See Appendix A for detailed subscale ratings.

### 3.2. Hypothesis 2: Did the Groups Differ in Terms of Brain Activation?

Hypothesis 2a proposed that both the lecture and body rhythm groups would exhibit greater brain activation than the control group during the post-test, with the body rhythm group showing the strongest activation in emotion-related regions. Mean HbO values for each ROI during pre-test and post-test phases were calculated for all groups under each condition, with higher HbO indicating greater activation (see Appendix A). ROI activation (β-values) was analyzed using FDR-corrected three-way repeated-measures ANOVA. Results are illustrated in Figure 4.

For lFPC activation, the interaction of time and teaching method was significant, *F*(2, 100) = 12.95, *p*_FDR_ < 0.001, *η_p_*^2^ = 0.21. Simple effects analysis revealed no significant pre-test differences between groups (*p* = 0.52). At post-test, the body rhythm showed significantly higher activation (0.02 ± 0.003) than both the control (−0.007 ± 0.003; *p* < 0.001) and lecture (−0.0002 ± 0.003; *p* < 0.001), consistent with Hypothesis 2a. No significant difference was found between lecture and control (*p* = 0.54). See Figure 5a.

For rFPC activation, the interaction of time and teaching method was significant, *F*(2, 100) = 7.52, *p*_FDR_ = 0.007, *η_p_*^2^ = 0.13. Simple effects analysis revealed no significant pre-test differences between groups (*p* = 0.08). At post-test, simple effects analyses revealed no significant (*p* = 1) post-test differences between the lecture (−0.04 ± 0.004) and the control (−0.002 ± 0.004), which is not consistent with Hypothesis 2a. Consistent with Hypothesis 2a, the body rhythm (0.019 ± 0.004) had significantly higher activation than the control (*p* = 0.001) and lecture (*p* < 0.001), as shown in Figure 5b.

Hypothesis 2a, which predicted that “post-test brain activation would be superior to the control group for both teaching methods,” received only partial support: The body rhythm group demonstrated significantly higher activation levels than the control group in both comparisons, consistent with previous research findings. This indicates that body rhythm training can enhance the intensity of neural responses during subsequent independent music processing. However, no superior post-test brain activation was observed in the lecture group compared to the control group. This result aligns with cognitive theory expectations—the transmission of declarative knowledge via lecture is prone to retrieval failure or forgetting outside the instructional context, leading to sub-significant neural activation. Notably, the prediction that “the body rhythm group outperforms the lecture group” was validated in both comparisons. These findings support the core tenet of embodied cognition theory: affective processing internalized through body rhythm more effectively induces sustained neuroplastic changes in the bilateral FPC brain region [30]—a hub for higher-order cognitive processing involving multi-cue integration—compared to externally mediated cognition via lecture teaching.

Hypothesis 2b predicted stronger teacher–student IBS in both lecture and body rhythm groups compared to controls, with body rhythm expected to show the highest IBS in emotion-/social-processing regions. IBS was calculated as [Teaching Stage—Baseline] within the 0.036–0.2 Hz band, with higher values indicating stronger teacher–student-task neural coupling. A 2 (method) × 2 (music) repeated-measures ANOVA with FDR correction was conducted on mean IBS values. See Appendix A for mean HbO by ROI.

For IBS on lFPC activation, the main effect of teaching method was significant, *F*(2, 100) = 4.16, *p*_FDR_ = 0.028, *η_p_*^2^ = 0.08; inconsistent with Hypothesis 2b, the difference in IBS between the lecture (−0.042 ± 0.015) and control (−0.015 ± 0.015) was not significant (*p* = 0.21). Consistent with Hypothesis 2b, the body rhythm (0.04 ± 0.015) had a significantly higher degree of IBS than control (*p* = 0.01) and lecture (*p* = 0.008). For IBS on lOFC activation, the main effect of teaching method was significant, *F*(2, 100) = 7.92, *p*_FDR_ = 0.01, *η_p_*^2^ = 0.14; inconsistent with Hypothesis 2b, the difference in IBS between the lecture (−0.023 ± 0.012) and control (−0.015 ± 0.012) was not significant (*p* = 0.67). Consistent with Hypothesis 2b, the body rhythm (0.022 ± 0.012) had a significantly higher degree of IBS than the control (*p* = 0.028) and lecture (*p* < 0.001). For IBS on rSTG activation, the main effect of teaching method was significant, *F*(2, 100) = 5, *p*_FDR_ = 0.04, *η_p_*^2^ = 0.09; inconsistent with Hypothesis 2b, the difference in IBS between the lecture (−0.003 ± 0.015) and control (−0.032 ± 0.016) was not significant (*p* = 0.19). The degree of difference in IBS between the lecture and the body rhythm (0.037 ± 0.015) did not reach the level of significance (*p* = 0.07). And consistent with hypothesis 2b, the degree of IBS was significantly higher in the body rhythm than the control (*p* = 0.002), See Figure 5c. See Appendix A for other results.

Hypothesis 2b was partially supported. While the body rhythm method significantly enhanced teacher-student interaction in the IBS compared to the control group, the lecture method did not. Critically, the body rhythm method induced stronger IBS than lecture teaching in multiple key social-emotional brain regions: the lOFC, associated with interoceptive emotional experience [32]; the lFPC, involved in multisensory integration and attentional maintenance [30]; and the rSTG, implicated in social auditory-motor integration and intent decoding [34]. These findings underscore that embodied rhythm teaching strengthens teacher-student neural coupling more effectively than didactic instruction, highlighting its role in facilitating emotionally resonant learning processes.

## 4. Discussion

### 4.1. Empirical Contributions

To our knowledge, this is the first study designed to investigate the effects of different teaching methods on emotional processing in music education. Using fNIRS hyperscanning in a realistic classroom scenario with 3 teachers and 103 students, we compared a body rhythm method, a lecture method, and a control group. Key findings are as follows:

First, consistent with Hypothesis 1a, both teaching methods were more effective than the control group in improving students’ valence ratings of positive music; however, only body rhythm teaching significantly enhanced emotional arousal, and its effect was significantly superior to that of lecture teaching. Additionally, it is noteworthy that the valence metric focuses on enhancing an individual’s valence discrimination ability. Superior emotional processing manifests as individuals perceiving greater pleasure from positive music while simultaneously demonstrating heightened sensitivity and precision in recognizing the negative emotional connotations of negative music. Therefore, lower negative valence scores here are interpreted as compelling evidence of improved perceptual accuracy rather than an indication of teaching failure. Second, the body rhythm teaching group reported significantly higher evaluations of teaching quality (Hypothesis 1b). At the neural level, only the body rhythm group showed significantly enhanced activation in the bilateral FPC during post-test music listening (Hypothesis 2a), and significantly stronger teacher–student IBS in emotion-related brain regions (lOFC, lFPC, rSTG) was observed exclusively during body rhythm teaching (Hypothesis 2b). Finally, correlation analysis between behavioral outcomes and brain outcomes indicates that, the body rhythm group recruited bilateral FPC resources more efficiently and synergistically when listening to positive music.

In summary, by integrating behavioral and fNIRS hyperscanning measures, this study demonstrates that embodied teaching—specifically, body rhythm pedagogy—enhances emotional engagement more effectively than traditional methods. This advantage is evidenced by heightened emotional arousal, increased teacher–student neural synchrony, and context-specific neural recruitment. Our findings establish a neurobehavioral basis for embodied learning and offer refined, mechanism-guided insights for optimizing music pedagogy.

### 4.2. Methodological Contributions

This study demonstrates the utility of fNIRS hyperscanning for capturing the brain dynamics of teacher–student interactions during emotional teaching. While the potential of neuroscience in education has been recognized since Thorndike (1913) [56], its application to affective pedagogy has remained limited [22]. Our work provides direct neurocognitive evidence on how teaching methods modulate both individual brain activation and teacher–student neural synchrony during music instruction, thereby enriching the empirical foundation for neuroscience-informed pedagogy.

### 4.3. Theoretical Implications

The first significant contribution of this study lies in examining the impact of different teaching methods on students’ emotional processing during music affect instruction, particularly in relation to embodied cognition and disembodied cognition. The lecture teaching method, rooted in meaningful learning theory [8] and widely employed for knowledge transmission [13,57], demonstrated a fundamental limitation: while it improved perceived emotional valence, it failed to enhance physiological arousal or elicit significant activation in emotion-related brain regions. This pattern underscores a critical constraint of traditional, disembodied pedagogy—it effectively transfers propositional knowledge about emotions but remains limited in evoking genuine emotional engagement or corresponding neural activity [58]. In contrast, the body rhythm teaching method, grounded in the SAME model’s core mechanisms of imitation and synchronization [16], facilitated embodied dynamic mapping of musical elements through improvisational movement. This approach transcended mere mirror neuron-mediated imitation [59], engaging higher-order cognitive processes such as metaphorical mapping through multisensory integration [60]. The empirical results validate this theoretical distinction: the body rhythm group demonstrated significant advantages in behavioral scores, neural activation intensity, and teaching quality evaluations compared to both lecture and control groups. This divergence strongly suggests that while the lecture method fosters a conceptual understanding of emotion, the body rhythm method enables a somatosensory experience of emotion, thereby offering a more direct pathway for emotional learning. This dissociation between knowing and feeling, clearly reflected in our behavioral and neural data, provides robust empirical support for embodied cognition theory. It demonstrates that knowledge about emotion and the lived experience of emotion are subserved by dissociable cognitive and neural pathways, which can be selectively engaged by different teaching methods.

Secondly, the study’s significant contribution lies in providing neurophysiological evidence for emotional teaching theory. According to the psychology of emotional teaching [22], teachers, students, and materials constitute the three core elements of emotional instruction, where the quality and strength of their interactions determine the effectiveness of the teaching. Our findings reveal that the body rhythm group exhibited significantly stronger teacher–student IBS in socio-emotional brain regions compared to both the lecture and control groups. This neural evidence demonstrates that body rhythm teaching more effectively integrates these three pedagogical elements, fostering “resonance” between teachers and students, thereby validating, at the neural level, the profound interpersonal connections emphasized by emotional teaching theory.

### 4.4. Practical Implications

This study’s findings provide important insights into the mechanisms of emotional processing and pedagogical optimization. The SAME model identifies imitation and synchronization as crucial mechanisms for music-induced emotion [16]. Our empirical results further demonstrate that the body rhythm teaching method, which closely aligns with these mechanisms, exhibits superior efficacy in music–emotion instruction, particularly in enhancing the intensity of emotional experience and promoting neural plasticity. When teachers employ pedagogical approaches that effectively engage imitation and synchronization mechanisms, they can significantly activate students’ relevant neural and cognitive resources during music–emotion processing, thereby elevating their overall emotional perception and experience. More importantly, this study reveals a potentially universal teaching mechanism: by mobilizing the fundamental social learning mechanism of imitation and synchronization, teaching methods can effectively activate students’ neural and cognitive resources during the learning process, thereby deepening their emotional and cognitive engagement. This finding applies not only to musical emotion processing but also holds significant implications for the theoretical development and paradigm innovation of general pedagogy. Therefore, its applications extend far beyond the musical domain (e.g., instrumental/vocal instruction, music therapy) and can directly inform and optimize any discipline requiring high emotional investment and social interaction, such as language arts or physical education. This research not only addresses the urgent need to enhance students’ emotional processing abilities in music education but also contributes a transferable core component for constructing an evidence-based, emotion-oriented interdisciplinary teaching framework.

### 4.5. Limitations and Future Directions

Despite being the first fNIRS hyperscanning investigation of emotional processing in interactive music teaching, this study has several limitations.

First, current research on body movement pedagogy primarily focuses on imitating fundamental musical elements (rhythm, dynamics, pitch, timbre) through localized body parts (e.g., fingers, arms), without systematically exploring how trunk, lower limb, or whole-body movements express deeper musical characteristics such as structure, harmony, and form. Future research should broaden the scope of bodily movements and musical parameters, refine the theoretical and practical framework of this method, and enhance its feasibility and applicability in authentic classroom settings. Furthermore, it is worth considering that the design of this study did not fully explore the potential of whole-body movement. This raises an open and thought-provoking empirical question: Would large-scale whole-body movements elicit stronger quantitative brain activation and generate neural patterns with qualitatively distinct characteristics compared to the localized movements employed here? Future research systematically comparing neural correlates across movements of varying intensity—from finger tapping to full-body dance—will be crucial for clarifying the body’s role in musical affect and establishing more precise “embodiment-brain activity” relationships.

Secondly, music-induced emotion involves multidimensional theoretical mechanisms. Juslin (2013) [61] proposed the BRECVEMA model, outlining seven mechanisms. Meyer’s (1956) [62] “expectancy theory” emphasized the role of musical structure expectations, later developed by Huron (2006) [63] into the five-stage Imagination–Tension–Prediction–Response–Appraisal (ITPRA) theory and others. These theories collectively form a multidimensional foundation for teaching music emotion, warranting further validation of mechanism-based instructional strategies in practical educational settings.

Third, in terms of experimental design, this study focuses on the traditional Chinese music system, providing empirical evidence for the effectiveness of body rhythm teaching and lecture teaching methods in emotional education within this specific context. Future research could further apply such teaching methods to diverse musical genres such as Western classical, jazz, and pop music, systematically examining their applicability and variations. This would reveal both common and culturally specific cognitive-neural mechanisms in musical emotional education, thereby advancing the development of a more inclusive theoretical framework for music education. Secondly, previous research has revealed significant age differences in musical emotion processing [64]. Future research should employ developmental designs to systematically examine the trajectory of teaching method effects across age groups, particularly comparing differences among children, adolescents, adults, and older adults. This will facilitate the exploration of individualized instructional approaches tailored to these distinct developmental stages. Such investigations will significantly advance the development of age-appropriate music education theories grounded in evidence from developmental cognitive neuroscience. Moreover, a key future direction involves exploring the role of teaching methods within audiovisual integrated environments. Building upon the auditory processing baseline established in this study, subsequent work will systematically introduce visual stimuli to examine whether teaching method effects transfer and modulate overall emotional processing constructed from multisensory inputs. This will significantly enhance the ecological validity and generalizability of research findings. Finally, although all teachers in this study underwent rigorous training and employed standardized protocols, the limited sample size (*n* = 3) precludes us from completely ruling out potential confounding effects of specific teaching styles on the results. Future research should validate the generalizability of these findings across larger and more diverse teacher samples, potentially employing multilevel models to account for teacher variation as random effects.

Fourth, in terms of measurement, emotional measurement is inherently multidimensional. Although this study combined subjective reports with fNIRS technology to provide complementary perspectives at behavioral and neural levels, this framework remains incomplete. Music-induced emotions are equally reflected in physiological responses (such as autonomic nervous activity) and external behaviors (such as facial expressions). Future research should focus on integrating multimodal technologies—for example, recording autonomic indicators via biofeedback devices and quantifying facial and bodily movements using motion capture systems—to cross-validate emotional responses across multiple levels and systematically uncover their underlying mechanisms. Furthermore, the teaching attitude and teaching method dimensions in the teaching quality evaluation only include one question. In future research, other scales may be considered. Finally, this study employed a “teaching-application” paradigm to examine how different teaching methods instantaneously shape students’ cognitive patterns for processing musical emotions. However, the research primarily revealed the initial formation and immediate application of this ability. Its long-term stability and dynamic consolidation process remain key issues requiring systematic exploration in future studies. Subsequent research should employ longitudinal tracking designs, conducting multiple assessments weeks or even months after intervention to map the complete trajectory of instructional effects over time. This will test whether the embodied cognitive mechanisms of imitation and synchronization—upon which the body rhythm teaching method relies—can genuinely foster deeper, more enduring neural and behavioral changes.

## 5. Conclusions

This study demonstrates that both lecture-based and body rhythm teaching methods enhance students’ emotional processing of music, with the latter showing broader effects. While lecture teaching improved emotional valence compared to controls, body rhythm teaching further elevated emotional arousal and overall experience. Neuroimaging revealed that the body rhythm group exhibited stronger activation in the bilateral FPC, and greater teacher–student neural synchrony in key social–emotional regions—including the OFC, lFPC, and rSTG. These findings suggest that embodying musical elements through movement enhances emotional engagement and promotes activity in emotion-processing neural networks.

## Figures and Tables

**Figure 1 brainsci-15-01253-f001:**
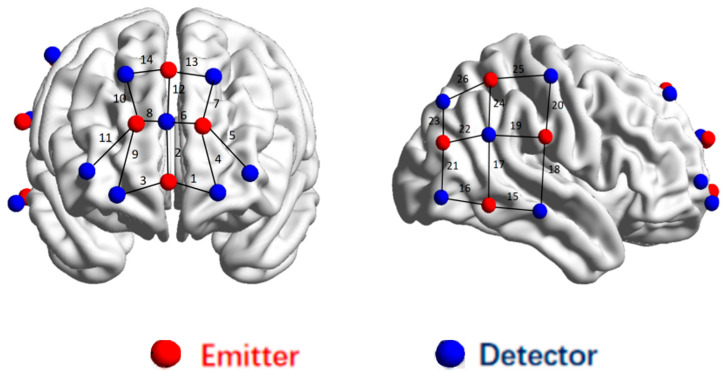
The optode probe set and channels on a 3D brain. Note: Red represents the transmitting optical pole, blue represents the receiving optical pole, and the numbers represent the channels.

**Figure 2 brainsci-15-01253-f002:**
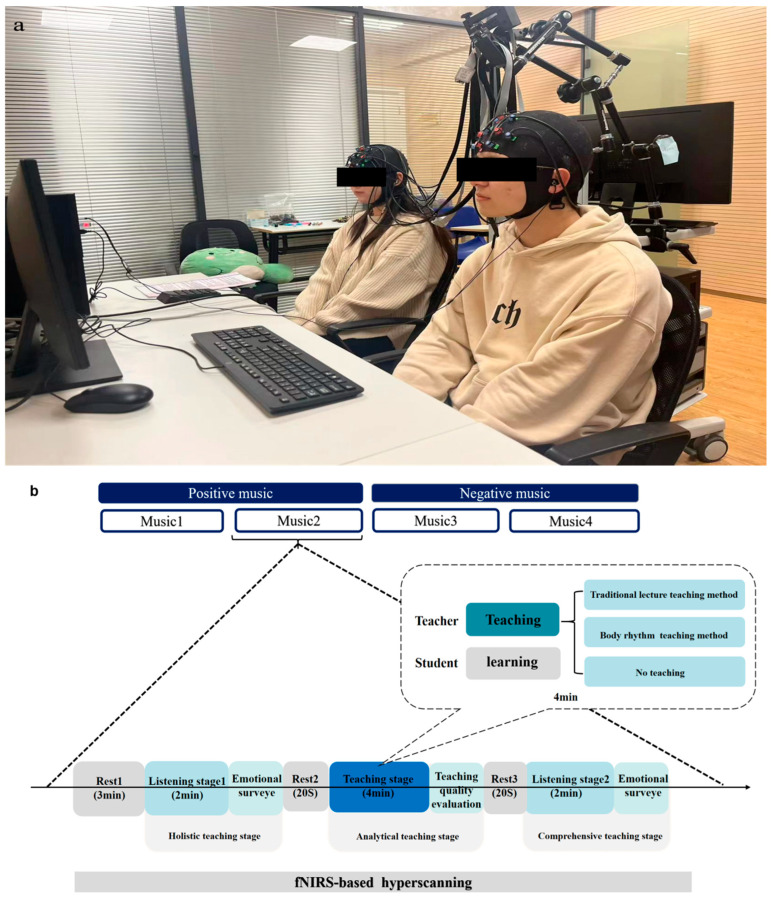
(**a**) Schematic diagram of the experimental scene. (**b**) Experimental flow of music appreciation class. (**c**) Specific design of teaching stage.

**Figure 3 brainsci-15-01253-f003:**
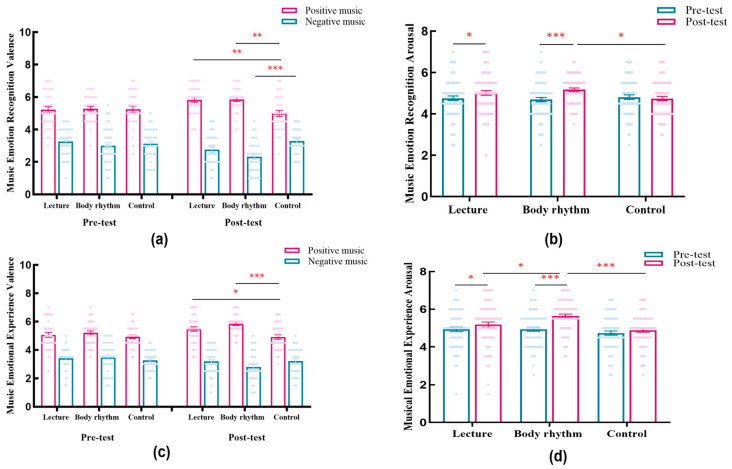
Music emotion processing results for the three groups (music emotion recognition valence (**a**), music emotion recognition arousal (**b**), music emotion experience valence (**c**), music emotion experience arousal (**d**)). Note: * *p* < 0.05; ** *p* < 0.01, *** *p* < 0.001.

**Figure 4 brainsci-15-01253-f004:**
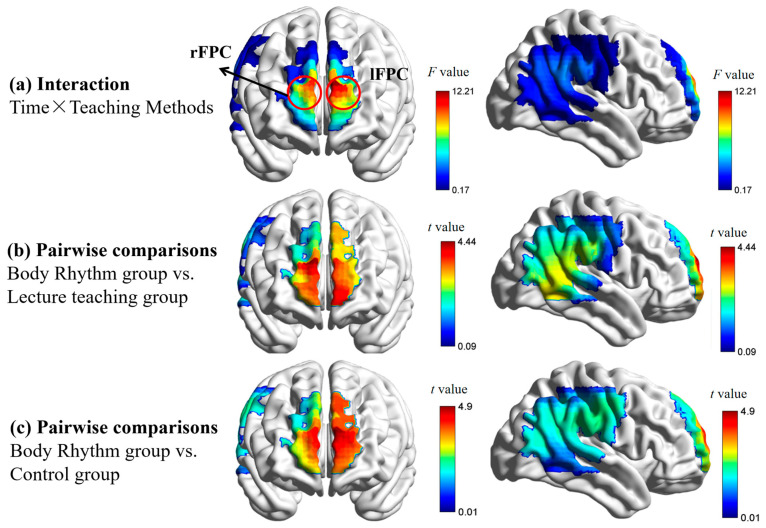
Results of analysis of variance. In (**a**), the F-value representing the interaction between time and teaching method indicates that the redder the color, the stronger the interaction in that brain region. (**b**,**c**) show the results of pairwise comparisons. (**b**) There was a significant difference between the post-test activation of the body rhythm teaching group and the lecture teaching group in the brain regions (lFPC, rFPC). (**c**) The post-test activation of the body rhythm teaching group showed significant differences in brain regions (lFPC, rFPC) compared to the control group.

**Figure 5 brainsci-15-01253-f005:**
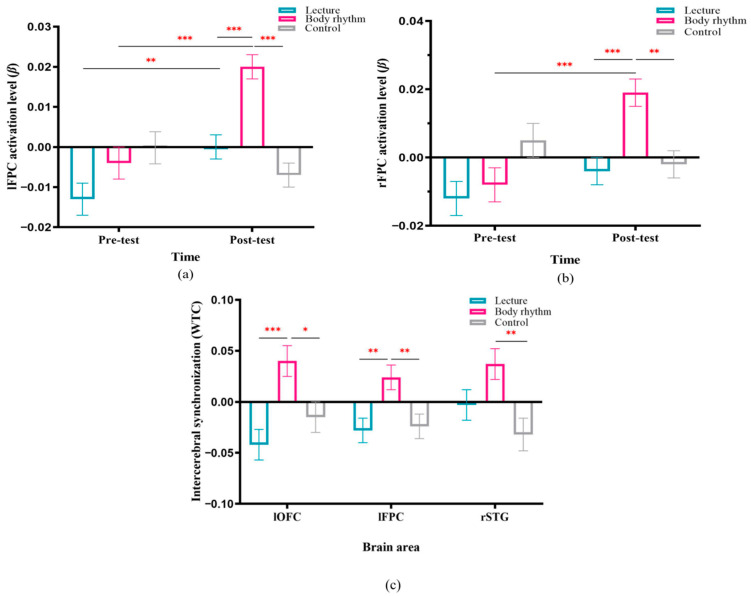
Graph of NIR imaging results. (**a**) Post-test of the activation level of the three groups on the lFPC during the music-listening phase. (**b**) Post-test activation levels on rFPC for the three groups during the music-listening phase. (**c**) Intercellular synchronization activity levels between teachers and students in lOFC, lFPC, and rSTG brain regions during the teaching phase. Note: * *p* < 0.05; ** *p* < 0.01, *** *p* < 0.001.

**Table 1 brainsci-15-01253-t001:** ROIs and corresponding fNIRS channels.

Brain Areas	ROI	Channel	Brain Areas	ROI	Channel
FPC	lBA10	2, 4, 5, 6	ANG	rBA39	20, 24, 25
	rBA10	8, 9, 11	SMG	rBA40	17, 18, 19
OFC	lBA11	1	ITG	rBA20	21
	rBA11	3	TPGmid	rBA21	22
dlPFC	lBA9	7, 12, 13	FFG	rBA37	23
	rBA9	10, 14	V3	rBA19	26
SI	rBA1	15	STG	rBA22	16

**Table 2 brainsci-15-01253-t002:** Mean and standard deviations for the dependent measures of three groups.

Dependent Variables		Lecture Teaching Group (*n* = 35)								Body Rhythm Teaching Group (*n* = 35)								Control Group (*n* = 33)							
		Pre-Test				Post-Test				Pre-Test				Post-Test				Pre-Test				Post-Test			
		Positive Music		Negative Music		Positive Music		Negative Music		Positive Music		Negative Music		Positive Music		Negative Music		Positive Music		Negative Music		Positive Music		Negative Music	
		M	SD	M	SD	M	SD	M	SD	M	SD	M	SD	M	SD	M	SD	M	SD	M	SD	M	SD	M	SD
**Musical Emotional Processing**	Emotion Recognition Valence	5.23	1.07	3.26	0.87	5.83	0.87	2.76	0.97	5.29	0.80	3.00	1.16	5.86	0.78	2.31	0.96	5.26	1.08	3.12	0.92	5.00	1.08	3.29	0.76
	Emotion Recognition Arousal	4.89	0.96	4.63	0.89	5.00	1.01	5.03	0.92	4.86	0.94	4.53	0.79	5.34	0.70	5.00	0.62	4.91	1.02	4.71	0.78	4.86	0.94	4.62	0.81
	Emotion Experience Valence	5.06	1.01	3.40	0.74	5.47	0.95	3.19	0.90	5.20	0.83	3.47	1.04	5.84	0.66	2.79	1.04	4.92	0.75	3.26	0.67	4.91	0.93	3.21	0.94
	Emotion Experience Arousal	5.19	0.96	4.67	0.96	5.27	1.11	5.10	1.22	5.09	0.98	4.79	0.81	5.91	0.75	5.37	0.83	4.85	0.83	4.62	1.02	5.05	0.88	4.73	0.84
**Teaching Quality Evaluation**						23.53	3.58	22.96	3.04					25.89	2.03	25.00	2.79					-	-	-	-

## Data Availability

The data presented in this study are openly available in Zenodo (https://zenodo.org) at https://doi.org/10.5281/zenodo.17089533 (accessed on 4 October 2025).

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
