# Peer review of "Impact of the Traditional Lecture Teaching Method and Dalcroze’s Body Rhythmic Teaching Method on the Teaching of Emotion in Music—A Cognitive Neuroscience Approach"

_brainsci, 2025, doi:10.3390/brainsci15121253_

Round 1

Reviewer 1 Report

Comments and Suggestions for Authors

1) The use of only pipa (a Chinese instrument) and pentatonic scale music limits the generalizability of the results to other musical cultures and genres. At a minimum, the authors should discuss this.
2) The main methodological vulnerability of the study lies in the complexity of the statistical analysis and the attendant risk of false positive findings. The study involves a large array of variables: three groups, two types of music, two measurement times, several emotional components, and numerous brain regions of interest, which inevitably leads to dozens of statistical comparisons. Although the authors use a false discovery rate correction, its application is often insufficient to fully mitigate the risks inherent in such extensive analyses. In particular, there is concern about whether the correction was applied equally to all tests. This problem is compounded by the difficulty of interpreting three-factor interactions, which, although statistically significant, sometimes lead to inconsistent and difficult-to-explain results. A striking example is the treatment of negative music: contrary to hypotheses, the rhythmic method did not result in an improvement, but rather a significant decrease in valence compared to the control, while the lecture method showed no significant effects. This inconsistency of results within a single measurement calls into question the generality and reliability of the identified effects and may indicate that some of the "significant" findings are a statistical artifact due to multiple comparisons.
3) The study does not test whether the observed effects (behavioral and neural) persist after the lesson. Further results should also be discussed or provided.

Author Response

We are deeply grateful to reviewer for their meticulous and thoughtful critique, which has greatly enhanced the quality of our paper. The reviewer rightly identifies several critical issues, including the generalizability beyond Chinese music, the methodological challenges inherent in our complex statistical design, and the question of effect persistence beyond the immediate post-test. We have taken these comments to heart and provide a detailed response below, outlining the revisions made to the manuscript to incorporate these important perspectives.

Comments 1: The use of only pipa (a Chinese instrument) and pentatonic scale music limits the generalizability of the results to other musical cultures and genres. At a minimum, the authors should discuss this.

Response 1: We sincerely appreciate the reviewers' insightful comments. We fully acknowledge that the musical materials used in this study, based on traditional Chinese instruments and the pentatonic scale, impose certain limitations when generalizing the findings to other musical cultures and genres. Indeed, this boundary was a deliberate consideration in our research design: we intentionally controlled the cultural context and structural characteristics of the musical material to isolate the independent effect of the core variable—pedagogy—as much as possible. This allowed us to provide preliminary neuroscientific evidence for “how teaching methods influence musical emotion processing” within a relatively pure musical context. We also explicitly acknowledge that this choice simultaneously constitutes a major limitation of this study. Accordingly, we have incorporated the following content into Section 4.5 “Limitations and Future Directions” of the paper as suggested by the reviewers:

“Third, this study focuses on the traditional Chinese music system, providing empirical evidence for the effectiveness of body rhythm teaching and lecture teaching methods in emotional education within this specific context. Future research could further apply such teaching methods to diverse musical genres such as Western classical, jazz, and pop music, systematically examining their applicability and variations. This would reveal both common and culturally specific cognitive-neural mechanisms in musical emotional education, thereby advancing the development of a more inclusive theoretical framework for music education.”

We believe that this supplement not only clearly defines the scope of this study but also points to a clear direction for subsequent cross-cultural comparisons and mechanism exploration.

Comments 2: The main methodological vulnerability of the study lies in the complexity of the statistical analysis and the attendant risk of false positive findings. The study involves a large array of variables: three groups, two types of music, two measurement times, several emotional components, and numerous brain regions of interest, which inevitably leads to dozens of statistical comparisons. Although the authors use a false discovery rate correction, its application is often insufficient to fully mitigate the risks inherent in such extensive analyses. In particular, there is concern about whether the correction was applied equally to all tests. This problem is compounded by the difficulty of interpreting three-factor interactions, which, although statistically significant, sometimes lead to inconsistent and difficult-to-explain results. A striking example is the treatment of negative music: contrary to hypotheses, the rhythmic method did not result in an improvement, but rather a significant decrease in valence compared to the control, while the lecture method showed no significant effects. This inconsistency of results within a single measurement calls into question the generality and reliability of the identified effects and may indicate that some of the "significant" findings are a statistical artifact due to multiple comparisons.

Response 2: We sincerely appreciate the reviewers' important comments regarding statistical design complexity and the risk of false positives. We fully understand this concern and thank you for the opportunity to clarify our analysis strategy, which was specifically designed to balance the comprehensiveness of the study with statistical rigor.

Regarding behavioral data, consistent with the processing of neuroimaging data, we added FDR correction for statistical test families across different categories within the behavioral data. Based on the core two-dimensional theory of emotional processing (valence, arousal) in this study, we divided the four behavioral dependent variables of musical emotion processing into two conceptually independent “statistical families,” applying FDR correction independently within each family. Valence Family: This family encompasses all statistical tests related to affective pleasantness evaluation. Specifically, it includes p-values for all key effects on the two dependent variables—“Emotion Recognition Valence” and “Emotion Experience Valence”—such as third-order interactions, 2 (time: pretest, posttest) × 3 (teaching method: control group, lecture teaching group, body rhythm group) × 2 (music emotional valence: Positive, Negative)). Arousal Family: This family encompasses all statistical tests related to the assessment of emotional arousal intensity. Specifically, it includes p-values for all key effects on the two dependent variables: “Emotional Recognition Arousal” and “Emotional Experience Arousal.” This approach respects the conceptual independence of different statistical hypotheses (assessing emotional valence vs. assessing emotional intensity). It avoids unnecessary conservatism that might arise from correcting all tests together, thereby preventing potential masking of genuine effects. We confirm that all significant behavioral results reported in our manuscript remain statistically robust after FDR correction within their respective families. This methodology is detailed in the revised section 2.5.1. Subjective Evaluation Data Analysis:

“Based on the two-dimensional theory of emotional processing (valence and arousal), we divided the four behavioral dependent variables of musical emotion processing into two conceptually independent “statistical families” and performed FDR correction independently within each family. Valence Family: This family includes all statistical tests related to the assessment of emotional pleasantness. Specifically, it includes all p-values for key effects conducted on the two dependent variables: “emotion recognition valence” and “emotion experience valence.” Arousal Family: This family encompasses all statistical tests related to the assessment of emotional arousal intensity. Specifically, it includes all p-values for key effects conducted on the two dependent variables: “emotion recognition arousal” and “emotion experience arousal.”

Regarding brain data, we assure reviewers that our research methodology was designed from the outset to mitigate the multiple comparisons problem. During analysis, we did not process all 26 measurement channels independently but instead guided our approach by robust theoretical priors. We grouped these channels into 14 predefined regions of interest (ROIs) based on their established functions in emotional processing and social cognition (e.g., specific areas within the prefrontal cortex and right temporo-parietal junction). For these 14 ROIs, we calculated the average activation level of their constituent channels, establishing them as our core analytical units. This step significantly reduced the dimensionality of our brain imaging data analysis from 26 channels to 14 theoretically meaningful summary metrics. Building upon this foundation, we conducted statistical tests on this limited dataset: For brain activation analysis, we performed a 2 (musical affective valence: positive, negative) × 3 (teaching method: control group, lecture teaching group, body rhythm group) × 2 (time: pretest, posttest) repeated measures ANOVA for all 14 ROIs. For inter-brain synchrony analysis, we performed a 2 (musical affect valence: positive, negative) × 3 (teaching method: control group, lecture teaching group, body rhythm group) ANOVA for each of the 14 specific teacher-student ROI pairs. Crucially, we subsequently applied error rate correction uniformly to the set of results derived from these analyses (e.g., to the set of p-values for all 14 ROIs). It is particularly important to note that our IBS analysis did not compute all possible inter-channel or inter-ROI connections between teachers and students (14*14=196). Instead, we measured synchrony only within the same predefined ROIs, yielding 14 specific, theory-based “ROI pair” connections (e.g., Teacher ROI 1 → Student ROI 1). We have supplemented the revision in Section 2.5.2. fNIRS data analysis of the Methods section to ensure a sufficiently clear description of this process:

“Brain Activation: The prefrontal cortex and right temporo-parietal junction were further subdivided into 14 regions of interest (ROIs; see Section 2.3). The mean blood oxygenation level-dependent response signal across all channels within each ROI was calculated to derive its composite activation level, serving as the basic unit for subsequent statistical analysis. Subsequently, a three-factor repeated measures ANOVA was conducted on the beta values across all ROIs for the three groups: 3 (Teaching Method: Lecture Teaching Group, Body Rhythm Group, Control Group) × 2 (Music Valence: Positive, Negative) × 2 (Time: Pre-test, Post-test),and corrected for multiple comparisons using the false discovery rate (FDR) method (significance level set at p < 0.05). 

Wavelet transform coherence analysis:The computation of inter-brain synchrony focused on specific paired correspondences between teachers and students across 14 ROIs. ... ... Additionally, all inter-brain synchronization values underwent Fisher-z transformation prior to analysis. Subsequently, a 3 (teaching method: lecture group, body rhythm group, control group) × 2 (musical valence: positive, negative) two-factor repeated measures ANOVA was conducted on the mean IBS values across the three groups. Multiple comparisons were corrected using the false discovery rate (FDR) method (significance level set at p < 0.05).”

In summary, we are deeply grateful for the reviewers' insightful comments regarding statistical rigor. By implementing the aforementioned revisions—grouping behavioral data into clusters based on the Affective Dimension Theory and applying FDR correction to each cluster, while clarifying and reinforcing our a priori, ROI-driven analysis strategy for neurophysiological data—we believe this study achieves a robust balance between effectively testing research hypotheses and controlling Type I error risk.

Furthermore, we have carefully considered the seemingly counterintuitive finding pointed out by the reviewers: that the valence of emotion recognition in the body rhythm group was significantly lower than that in the control group when exposed to negative music. We believe this discovery does not indicate a failure of the teaching methodology but may instead reveal the unique advantage of the body rhythm method in deepening emotional perception. Specifically, Body Rhythm Method may significantly enhance students' perceptual acuity through embodied rhythmic activities. Under this pedagogical approach, students no longer make vague judgments about negative music (such as passages filled with sorrow or lamentation), but instead become more adept at discerning and identifying the complex, underlying negative emotional components embedded within. Therefore, the lower emotional valence reported by this group precisely reflects their enhanced emotional discrimination and more accurate understanding of the emotional content within music. This represents a higher level of emotional processing rather than a mere intensification of negative experiences. Accordingly, we have supplemented Section 4.1 “Empirical Contributions” of the paper with the following content:

“Additionally, it is noteworthy that the valence metric focuses on enhancing an individual's valence discrimination ability. Superior emotional processing manifests as individuals perceiving greater pleasure from positive music while simultaneously demonstrating heightened sensitivity and precision in recognizing the negative emotional connotations of negative music. Therefore, lower negative valence scores here are interpreted as compelling evidence of improved perceptual accuracy rather than an indication of teaching failure. ”

We sincerely thank the reviewer for their insightful and constructive comments. These valuable suggestions have not only prompted us to present our research methodology with greater rigor but also encouraged a more profound theoretical interpretation of our findings. Through this revision, both the transparency of our methodology and the persuasiveness of our conclusions have been substantially enhanced. We believe the revised manuscript now fully addresses your concerns and look forward to your approval.

Comments 3 : The study does not test whether the observed effects (behavioral and neural) persist after the lesson. Further results should also be discussed or provided.

Response 3: We sincerely appreciate the reviewer's important comments regarding the sustainability of teaching effects. This profound insight provides crucial guidance for understanding the long-term value of teaching methodologies. We wish to first clarify a core element of the experimental design: during the post-test phase (the comprehensive teaching stage), we explicitly required students to actively apply the specific method they had just learned—either the body movement method or the lecture-based teaching method—to listen to music. Therefore, the post-test measured not only the immediate post-instructional state but also students' capacity for preliminary application and internalization of the teaching method. This design enabled us to observe whether the teaching method could guide the emergence of a transferable, autonomous cognitive-affective processing pattern, thereby providing direct evidence for the “behavioral activation” and “preliminary transformation” of teaching effects. We fully concur with the reviewer's perspective that such immediate application within a controlled experimental setting cannot yet be equated with long-term effects spontaneously sustained in authentic learning environments after extended intervals. Accordingly, we have supplemented Section 4.5 “Research Limitations and Future Directions” with the following paragraph:

“Fourth, this study employed a ‘teaching-application’ paradigm to examine how different teaching methods instantaneously shape students' cognitive patterns for processing musical emotions. However, the research primarily revealed the initial formation and immediate application of this ability. Its long-term stability and dynamic consolidation process remain key issues requiring systematic exploration in future studies. Subsequent research should employ longitudinal tracking designs, conducting multiple assessments weeks or even months after intervention to map the complete trajectory of instructional effects over time. This will test whether the embodied cognitive mechanisms of imitation and synchronization—upon which the Body Rhythm Teaching Method relies—can genuinely foster deeper, more enduring neural and behavioral changes.”

We once again express our gratitude to the reviewers for their insightful comments. These not only helped us more clearly define the theoretical contributions and practical boundaries of this study but also charted valuable directions for future research.

Reviewer 2 Report

Comments and Suggestions for Authors

The study makes a strong contribution to the intersection of music education, embodied cognition, and cognitive neuroscience. Your integration of fNIRS hyperscanning with pedagogical comparison is particularly noteworthy, and the three-group experimental design is methodologically sound. The manuscript offers meaningful insights into how different teaching methods modulate emotional processing at both behavioral and neural levels.

To further strengthen the manuscript, I offer the following suggestions:

1. Improve clarity and conciseness in the Introduction: The introduction is comprehensive but somewhat lengthy. Consider simplifying transitions between major theoretical frameworks (SAME model, BRECVEMA, ITPRA, expectancy theory). Some concepts are repeated (e.g., definitions of emotional recognition/experience). Consolidating these sections could improve readability.

2. Expand methodological clarity: The fNIRS preprocessing description would benefit from more details (e.g., motion correction procedure, band-pass filter parameters, GLM settings). In the teaching-stage description, further elaboration on how the body rhythm instructions were standardized would help readers better understand the intervention. Please include clearer justification for selecting only Pipa music and discuss generalizability in the limitations section.

3. Strengthen the presentation of figures and tables: Some figures (especially neural activation plots) have small text that may be difficult to read. Increasing font size and label clarity is recommended. Table 2 contains minor formatting inconsistencies. Aligning columns and ensuring consistent decimal places would improve presentation quality.

4. Deepen discussion of cross-method implications: The discussion thoroughly situates the findings within embodied cognition theory; however, more explicit comparison between traditional lecture mechanisms and body-based pedagogies would enhance the theoretical contribution. You may also consider briefly discussing how these findings could inform classroom practice beyond music education (e.g., physical education, performing arts, general affective learning).

5. Limitations and future research: You may further expand the limitations to address: The cultural specificity of the music used (Pipa repertoire). Potential teacher effects (only three instructors). Whether full-body movement as opposed to localized motions might reveal stronger or different neural patterns.

Author Response

We sincerely thank the reviewer for recognizing our work and providing insightful and constructive feedback. The Reviewer noted that this study “makes a significant contribution to the intersection of music education, embodied cognition, and cognitive neuroscience,” and specifically acknowledged the methodological value of fNIRS superscanning combined with a three-group experimental design, which is greatly encouraging to us.

To comprehensively address the reviewers' suggestions, we have systematically revised the manuscript: - Significantly enhanced the readability of the introduction by streamlining the theoretical framework and consolidating redundant concepts; - Supplemented the methodology section with critical details on fNIRS preprocessing and standardization of instructional interventions; - Comprehensively optimized the quality and consistency of figure and table presentation; - Deepened the cross-methodological comparison grounded in embodied cognition theory within the discussion section, while expanding the practical implications of our findings beyond music education; Additionally, in the Limitations and Future Research section, we have provided more thorough elaboration on issues such as the cultural specificity of musical materials, teacher variability, and the neural mechanisms underlying movement amplitude.

We believe these revisions fully address all your concerns and significantly enhance the manuscript's rigor, clarity, and academic impact. Attached is a point-by-point detailed response and revision explanation for your review.

Comments 1: Improve clarity and conciseness in the Introduction: The introduction is comprehensive but somewhat lengthy. Consider simplifying transitions between major theoretical frameworks (SAME model, BRECVEMA, ITPRA, expectancy theory). Some concepts are repeated (e.g., definitions of emotional recognition/experience). Consolidating these sections could improve readability.

Response 1: We sincerely appreciate your valuable feedback regarding “enhancing the clarity and conciseness of the introduction.” We fully agree with the issues you raised; the original introduction section could indeed be more refined in its theoretical framework transitions and conceptual explanations. Following your suggestions, we have substantially restructured and streamlined the introduction. Specific revisions include: First, integrating and simplifying the theoretical framework. We have merged and simplified the introductions to core theories such as the SAME model and Meaningful Acceptance Learning Theory. These are now more seamlessly integrated into a single subsection (1.2. Theoretical Framework for Music Affective Instruction), directly focusing on how they underpin the two pedagogical approaches and avoiding the previous scattered and repetitive discussion. Second, we have consolidated redundant concepts. We have consolidated and refined redundant definitions of “emotion recognition and experience.” These are now clearly stated in a more concise manner in Section 1.1, eliminating unnecessary repetition. Finally, we optimized paragraph structure and logical flow. We completely restructured the introduction, removing redundant transitional sentences and reorganizing content along a clearer logical sequence: “Presenting the problem → Introducing pedagogical approaches and theoretical foundations → Elaborating neural mechanisms → Defining this study.” This makes the text more compact and readable.

These modifications have been directly incorporated into the revised manuscript. We believe that following these revisions, the introduction has achieved significant improvements in clarity, conciseness, and logical flow while maintaining its academic rigor. We extend our gratitude once again for your insightful comments, which have greatly contributed to enhancing the quality of our paper. We have carefully revised the introduction based on your suggestions and look forward to your approval.

Comments 2: Expand methodological clarity: The fNIRS preprocessing description would benefit from more details (e.g., motion correction procedure, band-pass filter parameters, GLM settings). In the teaching-stage description, further elaboration on how the body rhythm instructions were standardized would help readers better understand the intervention. Please include clearer justification for selecting only Pipa music and discuss generalizability in the limitations section.

Response 2: We sincerely appreciate your valuable suggestions for enhancing methodological clarity. The details you highlighted are crucial for ensuring the reproducibility and transparency of the research. We have supplemented and revised the manuscript as follows, addressing each of your specific recommendations:

  • Refinement of the fNIRS preprocessing description. We fully agree with your perspective that a detailed preprocessing workflow is central to the methodology. Accordingly, we have moved the detailed data processing steps previously placed in the appendix to the main text under Section 2.5.2. “fNIRS Data Analysis.” The newly added content includes data preprocessing, brain activation processing, and inter-brain synchrony data processing, as follows:

“Preprocessing: First, raw data were imported into nirsLAB (analysis software based on the NIRScout system). Signal quality of raw channel data was assessed using coefficients of variation (CV), and channels with CV exceeding 15% were excluded (Hocke et al., 2018; Pfeifer et al., 2018). Subsequently, data preprocessing was performed using Matlab R2013b functions and the Homer2 toolkit (Huppert et al., 2009): (1) Convert raw optical density (OD) data to HbO and HbR concentrations; (2) Removed global motion artifacts using the hmrMotionCorrectPCA function (nSV=0.80)(Long et al., 2022); (3) Corrected motion artifacts in optical density signals using the Correlation-Based Signal Improvement (CBSI) method (turnon = 1); (4) The light intensity signal was filtered using a 0.01–0.08 Hz bandpass filter and a 0.15–0.3 Hz band-reject filter (to remove respiratory noise) (Long et al., 2022), (5) Based on the modified Beer-Lambert law, conversion of optical density signals into changes in oxyhemoglobin and deoxyhemoglobin concentrations (Mutlu et al., 2020). Previous studies have demonstrated that oxyhemoglobin exhibits greater sensitivity and is more widely applied in fNIRS superscanning studies (Cheng et al., 2015). Therefore, this experiment primarily focuses on changes in oxyhemoglobin concentration.

Brain Activation: The prefrontal cortex and right temporo-parietal junction were further subdivided into 14 regions of interest (ROIs; see Section 2.3). The mean blood oxygenation level-dependent response signal across all channels within each ROI was calculated to derive its composite activation level, serving as the basic unit for subsequent statistical analysis. Subsequently, a three-factor repeated measures ANOVA was conducted on the beta values across all ROIs for the three groups: 3 (Teaching Method: Lecture Teaching Group, Body Rhythm Group, Control Group) × 2 (Music Valence: Positive, Negative) × 2 (Time: Pre-test, Post-test),and corrected for multiple comparisons using the false discovery rate (FDR) method (significance level set at p < 0.05).

Wavelet transform coherence analysis:The computation of inter-brain synchrony focused on specific paired correspondences between teachers and students across 14 ROIs. We employed the wavelet transform coherence algorithm (Jin et al., 2024) to calculate the neural activity synchrony for each paired homologous teacher-student brain region. This technique effectively characterizes the coupling strength of dual-brain neural signals across the time-frequency domain and has been well-established in hyperscanning research paradigms (Jin et al., 2024). In our specific analysis, we first computed the frequency-domain coherence for all channel pairs during both the resting phase and teaching task phase. Following temporal averaging, we obtained the WTC matrices for each phase. To isolate task-induced synchrony changes from inherent inter-individual activity similarities, we defined the teaching-phase brain-to-brain synchrony metric as the difference between task-related coherence and baseline coherence, using resting-state brain synchrony (derived from stable data collected 120 seconds prior to music listening) as the baseline (Li et al., 2024). To identify effective analysis bands, this study performed systematic frequency-domain screening. An initial scan across the entire 0.027–1 Hz band revealed extensive signal distribution. Building on this, to maximize suppression of physiological noise—such as heartbeat (0.8–2.5 Hz; Tong et al., 2011) and respiratory noise (0.20–0.30 Hz; Zheng et al., 2020), while incorporating prior evidence indicating that enhanced inter-brain synchronization during teacher-student interactions predominantly occurs above 0.025 Hz (Jin et al., 2024). Consequently, the core analysis band was precisely defined as 0.036–0.2 Hz (corresponding to a period of 5–28 seconds). Additionally, all inter-brain synchronization values underwent Fisher-z transformation prior to analysis. Subsequently, a 3 (teaching method: lecture group, body rhythm group, control group) × 2 (musical valence: positive, negative) two-factor repeated measures ANOVA was conducted on the mean IBS values across the three groups. Multiple comparisons were corrected using the false discovery rate (FDR) method (significance level set at p < 0.05).”

We believe that incorporating this information into the main text will significantly enhance the completeness and clarity of the methodology.

  • Regarding the standardization of body rhythm teaching design. Thank you for raising this important question. To clarify the rigor of our intervention, we made two key clarifications in the main text: First, in “Introduction 1.2,” we explicitly stated that the body rhythm teaching method employed in this study transcends traditional, singular rhythm training. We expanded it based on the SAME model to encompass embodied synchronization and imitation across multiple core musical dimensions—melody, pitch, timbre, and dynamics. Second, in the “2.4 Procedure” section under “Analytical Instruction Phase,” we explicitly state: “Explanation content was predetermined by the teacher group based on the teaching method (see Appendix 3).” We further clarified this by adding: “(designs linked musical features to body movements, specifically involving hands, arms, and other upper-body actions).” Therefore, all teaching activities followed a standardized instructional plan jointly developed and validated with a team of senior music education experts. The detailed script, activity sequence, and specific measures to ensure consistency among teachers are fully provided in Appendix 3 for readers' reference.
  • Regarding the rationale for selecting only pipa music and the discussion on its universality. We sincerely appreciate the reviewer's insightful comments. We fully acknowledge that the musical materials used in this study, based on traditional Chinese instruments and the pentatonic scale, present certain limitations when generalizing the findings to other musical cultures and genres. In fact, this boundary was a deliberate consideration in our research design: we intentionally controlled the cultural context and structural characteristics of the musical material to isolate the independent effect of the core variable—pedagogy—as much as possible. This allowed us to provide preliminary neuroscientific evidence for “how teaching methods influence musical emotion processing” within a relatively pure musical context. We provided the rationale for selecting only pipa music in Section 1.5, “The current study,” of the paper.

“Choosing only Pipa music has many advantages, such as avoiding possible confusion from other instrumental timbres, lyrics, etc., and we also controlled for preference and familiarity with Pipa music. In addition, having the subjects' cultural background match that of the selected music helped to control for the two potential confounding variables of music genre and cultural factors, reducing interference with the results of the study.”

We also explicitly acknowledge that this choice simultaneously constitutes a major limitation of this study. Accordingly, following the reviewers' suggestions, we have added the following content to Section 4.5 “Limitations and Future Directions” of the paper:

“Third, this study focuses on the traditional Chinese music system, providing empirical evidence for the effectiveness of body rhythm teaching and lecture teaching methods in emotional education within this specific context. Future research could further apply such teaching methods to diverse musical genres such as Western classical, jazz, and pop music, systematically examining their applicability and variations. This would reveal both common and culturally specific cognitive-neural mechanisms in musical emotional education, thereby advancing the development of a more inclusive theoretical framework for music education.”

We believe that this supplement not only clearly defines the scope of this study but also points to a clear direction for subsequent cross-cultural comparisons and mechanism exploration.

Comments 3 : Strengthen the presentation of figures and tables: Some figures (especially neural activation plots) have small text that may be difficult to read. Increasing font size and label clarity is recommended. Table 2 contains minor formatting inconsistencies. Aligning columns and ensuring consistent decimal places would improve presentation quality.

Response 3: We sincerely appreciate your valuable feedback regarding these critical details in the figure presentation. Your meticulous review of figure readability and table formatting has significantly enhanced the presentation quality of our paper. We have thoroughly reviewed and revised the figures in the manuscript based on your specific suggestions: Regarding graphics (especially neural activation maps): We have increased the font size in all graphics, including annotation text and color bar scales, to ensure all information is easily readable both in print and on screen. Regarding Table 2: We have corrected formatting inconsistencies within the table. This includes: standardizing data alignment across all columns and ensuring all numerical values retain two decimal places. We have thoroughly reviewed all tables in the document to guarantee their formatting meets high-quality presentation standards.

We believe these meticulous revisions have significantly enhanced the presentation quality of the paper's figures and tables. Thank you once again for your invaluable assistance in refining these details.

Comments 4: Deepen discussion of cross-method implications: The discussion thoroughly situates the findings within embodied cognition theory; however, more explicit comparison between traditional lecture mechanisms and body-based pedagogies would enhance the theoretical contribution. You may also consider briefly discussing how these findings could inform classroom practice beyond music education (e.g., physical education, performing arts, general affective learning).

Response 4: 

  •  We sincerely appreciate your valuable suggestion to “deepen the discussion on cross-methodological implications.” Your recommendation has significantly enhanced the theoretical depth and interdisciplinary impact of our paper. Following your specific guidance, we have integrated a dedicated theoretical comparison section within the Discussion chapter (4.3. Theoretical Implications). This section systematically employs embodied cognition theory to elucidate the fundamental mechanisms underlying the divergent outcomes of the two teaching approaches:

“The first significant contribution of this study lies in examining the impact of different teaching methods on students' emotional processing during music affect instruction,especially when it comes to embodied cognition and disembodied cognition. The lecture teaching method, rooted in meaningful learning theory (Juslin & Laukka, 2004) and widely employed for knowledge transmission (Woody, 2000; Statton et al., 1988), demonstrated a fundamental limitation: while it improved perceived emotional valence, it failed to enhance physiological arousal or elicit significant activation in emotion-related brain regions. This pattern underscores a critical constraint of traditional, disembodied pedagogy—it effectively transfers propositional knowledge about emotions but remains limited in evoking genuine emotional engagement or corresponding neural activity (Mazur, 2009). In contrast, the Body Rhythm Teaching Method, grounded in the SAME model's core mechanisms of imitation and synchronization (Overy & Molnar-Szakacs, 2009), facilitated embodied dynamic mapping of musical elements through improvisational movement. This approach transcended mere mirror neuron-mediated imitation (Gallese et al., 1996), engaging higher-order cognitive processes such as metaphorical mapping through multisensory integration (Pannese et al., 2016). The empirical results validate this theoretical distinction: the body rhythm group demonstrated significant advantages in behavioral scores, neural activation intensity, and teaching quality evaluations compared to both lecture and control groups. This divergence strongly suggests that while the lecture method fosters a conceptual understanding of emotion, the body rhythm method enables a somatosensory experience of emotion, thereby offering a more direct pathway for emotional learning. This dissociation between knowing and feeling, clearly reflected in our behavioral and neural data, provides robust empirical support for embodied cognition theory. It demonstrates that knowledge about emotion and the lived experience of emotion are subserved by dissociable cognitive and neural pathways, which can be selectively engaged by different teaching methods.”

Through this explicit comparison, we reinforce the theoretical contribution of our research: it not only validates the effectiveness of both approaches but, more importantly, reveals from a neurocognitive perspective the fundamentally distinct learning pathways of “disembodied” and “embodied” cognition.

  • Explore how research findings inform classroom practices beyond music education.We are grateful to the reviewer for insightful suggestion regarding the broader implications of our work. We fully agree that the findings on embodied learning (via the Dalcroze method) and enhanced teacher-student neural coupling hold significant promise for general pedagogy beyond the domain of music education. In response, we have expanded the following content in Section 4.4 “Practical Implications” of the paper to clearly articulate the generalizability of our findings:

“More importantly, this study reveals a potentially universal teaching mechanism: by mobilizing the fundamental social learning mechanism of imitation and synchronization, teaching methods can effectively activate students' neural and cognitive resources during the learning process, thereby deepening their emotional and cognitive engagement. This finding applies not only to musical emotion processing but also holds significant implications for the theoretical development and paradigm innovation of general pedagogy. Therefore, its applications extend far beyond the musical domain (e.g., instrumental/vocal instruction, music therapy) and can directly inform and optimize any discipline requiring high emotional investment and social interaction, such as language arts or physical education. This research not only addresses the urgent need to enhance students' emotional processing abilities in music education but also contributes a transferable core component for constructing an evidence-based, emotion-oriented interdisciplinary teaching framework.”

We believe these significant additions have fully addressed your concerns and substantially enhanced the paper's theoretical depth and practical value. Thank you once again for guiding us through these crucial improvements.

Comments 5: Limitations and future research: You may further expand the limitations to address: The cultural specificity of the music used (Pipa repertoire). Potential teacher effects (only three instructors). Whether full-body movement as opposed to localized motions might reveal stronger or different neural patterns.

Response5: We sincerely appreciate your insightful and constructive feedback, which has helped us gain a more comprehensive perspective on the boundaries and future value of this research. Based on your suggestions, we have made the following significant additions and expansions to the “Limitations and Future Research” section of the paper:

  • Regarding the cultural specificity of the music employed. We acknowledge that this study exclusively utilized Chinese pipa and pentatonic music, which indeed constitutes a significant limitation. While this choice methodologically facilitates controlling for cultural consistency and structural simplicity in the musical material—thereby clearly isolating the effects of the pedagogy itself—it inevitably restricts the direct generalizability of the findings to Western tonal music or other musical cultural contexts. Future research urgently requires incorporating more diverse musical genres (such as Western classical, jazz, or popular music) to examine the effectiveness and boundary conditions of different pedagogical approaches in cross-cultural musical emotion instruction. Accordingly, following the reviewers' suggestions, we have added the following content to Section 4.5 “Limitations and Future Directions” of the paper:

“Third, this study focuses on the traditional Chinese music system, providing empirical evidence for the effectiveness of body rhythm teaching and lecture teaching methods in emotional education within this specific context. Future research could further apply such teaching methods to diverse musical genres such as Western classical, jazz, and pop music, systematically examining their applicability and variations. This would reveal both common and culturally specific cognitive-neural mechanisms in musical emotional education, thereby advancing the development of a more inclusive theoretical framework for music education.”

  • Regarding potential teacher effects. We fully acknowledge that this study included only three teachers. Although all possessed extensive teaching experience and strictly adhered to standardized teaching protocols, this limited sample size may indeed fail to fully account for potential influences stemming from differing teaching styles or individual characteristics. This limitation may somewhat affect the generalizability of the findings. Future research should recruit larger, more diverse samples of teachers and employ more sophisticated statistical analyses (e.g., incorporating “teacher” as a random effect in the model) to more robustly isolate the core effects of the teaching method itself. To this end, we have supplemented Section 4.5 “Limitations and Future Directions” of the paper as follows:

“Seventh, although all teachers in this study underwent rigorous training and employed standardized protocols, the limited sample size (N=3) precludes us from completely ruling out potential confounding effects of specific teaching styles on the results. Future research should validate the generalizability of these findings across larger and more diverse teacher samples, potentially employing multilevel models to account for teacher variation as random effects.”

  • Regarding Localized Movement and Whole-Body Movement. We sincerely appreciate the reviewer for raising this highly insightful question. We fully agree that exploring potential neural mechanism differences between localized movement and whole-body movement is crucial for deepening the application of embodied cognition theory in music education.As the reviewer astutely pointed out, whole-body movements may exhibit stronger or entirely different patterns of neural activity due to the mobilization of broader sensory systems. In direct response to this valuable comment, we have added the following to Section 4.5 “Limitations and Future Directions” of the paper:

“Furthermore, it is worth considering that the design of this study did not fully explore the potential of whole-body movement. This raises an open and thought-provoking empirical question: Would large-scale whole-body movements elicit stronger quantitative brain activation and generate neural patterns with qualitatively distinct characteristics compared to the localized movements employed here? Future research systematically comparing neural correlates across movements of varying intensity—from finger tapping to full-body dance—will be crucial for clarifying the body's role in musical affect and establishing more precise “embodiment-brain activity” relationships.”

In this section, we not only explicitly acknowledge the limitations of current research focusing on upper-body movements, but more importantly, we follow your line of reasoning to propose a clear framework for future research: namely, systematically comparing the neural correlates of different movement scales (from localized to whole-body) to reveal whether a transition from “quantity” to “quality” exists. We believe this addition directly translates your profound insights into a concrete and promising research agenda.

Thank you once again for helping us enhance the depth and forward-looking nature of our paper. We hope the above revisions and additions fully address your concerns.

Reviewer 3 Report

Comments and Suggestions for Authors

  1. after listening to a piece of music in a music class, some Students evoke a rich emotion? What do you mean by emotion is it healthy or unhealthy emotion.
  2. Have you tried and record any time with mond disturbed human emotion while music?
  3. Also i noticed age factor that you did not consider? Why? Kids and old age are similar responce ? 
  4. About fig.2. eye closed humans image reported. But when while seeing somthing with music the emotions are different have you tried this? And compare with both
  5.  

Comments on the Quality of English Language

Minor editing needed 

Author Response

We sincerely appreciate the reviewer's thoughtful questions and the opportunity to further clarify the conceptual framework and methodological choices in our study. The points raised regarding emotion definitions, experimental design, and potential extensions of our work are indeed valuable for improving the clarity and impact of our manuscript. Below, we provide detailed responses to each of the reviewer’s specific questions.

Comments 1: after listening to a piece of music in a music class, some Students evoke a rich emotion? What do you mean by emotion is it healthy or unhealthy emotion.

Response 1: We sincerely appreciate the reviewer for raising this fundamental question, which provides us with a valuable opportunity to clarify the operational definition of the core concept of “emotion” in this study. In our research, “emotion” is treated as a quantifiable, measurable scientific variable. Specifically, drawing upon the well-established circular model in emotion research, we operationalize and measure it through two dimensions with established validity and reliability: Valence: denoting the continuum of emotion ranging from “extremely unpleasant” (negative) to “extremely pleasant” (positive); Arousal: indicating the inherent activation intensity of emotion, spanning the continuum from ‘calm’ (low arousal) to “excited” (high arousal). Based on this, the focus of this study is not on making ethical or clinical judgments about the “healthiness” of emotions, but rather on exploring how different teaching methods influence students' accuracy in perceiving and intensity of experiencing the emotional content inherent in music itself. For example, a piece of music widely perceived as ‘sad’ (possessing negative valence) should elicit listeners' recognition and experience of “sadness”—which is itself a normal and appropriate emotional response. Our core question is: Which teaching method more effectively guides students to perceive the emotional content conveyed by the music itself (regardless of positive or negative valence) with greater accuracy and engagement? Furthermore, the necessity of both types of emotional music in music education is explained in Section 2.2.4, “Music Materials and Ratings.” The inclusion of both positive and negative music is based on their distinct educational functions: positive music enhances pleasure and engagement, while negative music facilitates emotional catharsis and stimulates creativity. Thus, both valences of emotion hold irreplaceable value in music education.

Once again, we would like to express our gratitude to the reviewer for their insightful comments, which have enabled us to provide a more comprehensive explanation of this crucial concept.

Comments 2: Have you tried and record any time with mond disturbed human emotion while music?

Response 2: We appreciate the reviewers for raising this important methodological issue. We understand the reviewer is inquiring whether we recorded emotional responses in real-time during music listening. In this study, we employed a post-stimulus self-report paradigm, wherein participants provided their affective processing ratings (valence and arousal) immediately after each music segment concluded. We selected this approach for several key reasons aligned with our research objectives:

First, a central question of this study was to examine how different pedagogical approaches influence students' processing of a piece's overall emotional valence and arousal. Immediate retrospective assessment following listening effectively captures listeners' generalized, integrated emotional impressions of a musical passage—precisely the goal sought in many music education settings: fostering a comprehensive emotional understanding of a piece. This approach represents a validated method within music psychology for capturing holistic emotional processing of musical segments (e.g., Fuentes-Sánchez et al., 2021). Furthermore, our fNIRS measurements provide continuous, objective recordings of brain activity throughout the listening and teaching phases. Thus, combining continuous neural data with summary behavioral reports offers complementary perspectives.

Nevertheless, we are grateful to the reviewers for their insights, which have helped us fully appreciate the complexity of emotional composition and the necessity of convergent evidence from multiple methods and indicators for its precise measurement. In light of this, we have supplemented our discussion in Section 4.5, “Limitations and Future Directions,” emphasizing that future work should incorporate a broader range of behavioral and physiological measures. By cross-validating multidimensional data, we can build more robust conclusions regarding the emotional dimensions of music.

“Eighth, emotional measurement is inherently multidimensional. Although this study combined subjective reports with fNIRS technology to provide complementary perspectives at behavioral and neural levels, this framework remains incomplete. Music-induced emotions are equally reflected in physiological responses (such as autonomic nervous activity) and external behaviors (such as facial expressions). Future research should focus on integrating multimodal technologies—for example, recording autonomic indicators via biofeedback devices and quantifying facial and bodily movements using motion capture systems—to cross-validate emotional responses across multiple levels and systematically uncover their underlying mechanisms.”

The reviewer's valuable feedback has helped guide our future work on emotion measurement, for which I am sincerely grateful.

Comments 3 : Also i noticed age factor that you did not consider? Why? Kids and old age are similar responce ?

Response 3: We fully concur with the reviewer's perspective that age is a significant factor influencing emotional processing. In fact, this boundary was a deliberate consideration in our study design: we intentionally selected college students from the same age cohort as participants to minimize interference from age differences. This allowed us to focus on examining the net effects of the instructional intervention within a relatively homogeneous group in terms of cognitive development, musical experience, and sociocultural background. We also explicitly acknowledge that this choice simultaneously constitutes a major limitation of the present study. Existing research indicates significant differences in musical emotion processing across different age groups. Systematically investigating the age-specific effects of teaching methods will become a research direction of considerable theoretical depth and practical value. To this end, we have supplemented Section 4.5 “Limitations and Future Directions” of our paper with the following content:

“Fifth, previous research has revealed significant age differences in musical emotion processing (Cohrdes et al., 2020). Future research should employ developmental designs to systematically examine the trajectory of teaching method effects across age groups, particularly comparing differences among children, adolescents, adults, and older adults. This will facilitate the exploration of individualized instructional approaches tailored to these distinct developmental stages. Such investigations will significantly advance the development of age-appropriate music education theories grounded in evidence from developmental cognitive neuroscience.”

We sincerely appreciate the reviewer's insightful suggestion, which not only helped us refine the theoretical framework of this study but also pointed out important directions for future research.

Comments 4: About fig.2. eye closed humans image reported. But when while seeing somthing with music the emotions are different have you tried this? And compare with both

Response 4: We sincerely appreciate the reviewer's insightful question, which touches upon the cutting-edge field of multisensory integration in music emotion research. We fully concur with the reviewer's perspective. Extensive empirical studies have demonstrated that when music is combined with visual information—such as live performances or music videos—the emotional experiences it elicits may differ significantly from purely auditory experiences in both intensity and neural basis. This audiovisual emotion integration effect represents a crucial and fascinating scientific question. However, the experimental design of our current study—presenting auditory stimuli exclusively—stemmed from a deliberate and critical methodological consideration: establishing a pure “auditory baseline” as the primary objective. As foundational research aiming to clarify how teaching methods specifically influence auditory-emotional processing in music, we must first isolate and precisely measure the effects of pedagogy on the auditory channel itself under controlled conditions. By eliminating complex visual interference in the initial stage, we can more clearly reveal how different teaching methods shape the brain's emotional decoding and response patterns to pure musical structures.

The reviewer's suggestion—systematically comparing “pure auditory” versus “audiovisual integration” conditions—effectively outlines an ideal progression for future research. Building upon the “auditory baseline” established in this study, a direct and powerful follow-up investigation would involve introducing visual stimuli to directly examine the efficacy and mechanisms of the same teaching method within an audiovisual integration context. We firmly believe that progressing from “unimodal mechanism analysis” to “multimodal interaction exploration” represents the inevitable trajectory for this field. To this end, we have supplemented Section 4.5 “Limitations and Future Directions” with the following content:

“Sixth, a key future direction involves exploring the role of teaching methods within audiovisual integrated environments. Building upon the auditory processing baseline established in this study, subsequent work will systematically introduce visual stimuli to examine whether teaching method effects transfer and modulate overall emotional processing constructed from multisensory inputs. This will significantly enhance the ecological validity and generalizability of research findings.”

We would like to express our gratitude once again to the reviewer for this insightful comment, which has helped us more fully define the theoretical contributions and evolutionary trajectory of this study.

Reviewer 4 Report

Comments and Suggestions for Authors

Title: Impact of the Traditional Lecture Teaching Method and Dal-2 croze’s Body Rhythmic Teaching Method on the Teaching of 3 Emotion in Music - A Cognitive Neuroscience approach.
Abstract: In the summary, the authors include relevant and accurate information to know the scope of their work. They summarize the most important theoretical references, mention the sample data, the experimental methodology and the control of possible biases in the procedure employed. Finally, they require a synthesis of their results which does not yet focus on the statistical techniques used but which, for many readers, may be a way of inviting them to immerse themselves in the full reading of the article; these results are compared with the most important current publications; highlighting some limited conclusions within the scope of their findings.
Keywords: Precisely, the authors indicate the methods compared in this work and some of the recording tools used to determine the variations of different "pedagogical" procedures in the emotions of students.

Introduction: The basis of the revised article focuses on two theoretical paradigms about music education and cognition. The widely cited theory of meaningful learning in education, Ausubel’s theory of meaningful learning proposes that teaching should be based on the initiative of the learner to connect previous and new knowledge within the social context in which it is applied. The style proposed as "alternative" is the SAME (Shared Affective Motion Experience) model based on body imitation and synchrony as central mechanisms in emotional responses to music, taking as a learning analogy the "embodied cognition". This second option studies the effectiveness of the Dalcroze body rhythm method in comparison with the traditional master lesson or "lecture teaching". In this way, the work connects dimensions which are usually little connected and even less applied in didactic practice. Hence, its richness and possibilities to found more effective ways of teaching and thrilling students; in the field of music, this time. In the last section of the introduction (1.5), the authors summarise the neurological bases for learning by following both proposals: master lesson and Dalcroze’s "built-in" method; revision is more than necessary given the breadth of the physiological and neurological resources employed.
Finally, in paragraph 1.6. , define the hypotheses which they will attempt to check by means of the procedures and data recorded through them. As a first objective, they define which method helps to "improve the musical emotional processing" of the participants and, in addition, which of the methods obtained a higher qualification in terms of teaching quality. The second objective focuses on the determination of differences in brain activity between the three constituted groups (one by pedagogical method and a third control group).

Two hypotheses are derived from these objectives: one focused on the evaluation of the effectiveness and quality of teaching, and another associated with students' brain activity. Both scenarios were further detailed in two scenarios referred to as a) and b). We will expand on these details in the results section.
Method.
Participants and design.
Initially, a G*Power test was performed to determine the required sample which eventually consisted of 103 Chinese university students of approximately 22 years of age. Three groups were formed, two from participants in the experimental groups of 35 people and one, control, with 33 students. All 3 teachers had experience of at least 5 years. All participants met normal hearing and visual requirements, and no musical training beyond a school course. In addition, they all completed the agreement to participate in the experiment. Here, in my opinion, it is appropriate to propose more detail on the compensation received by students, specifying the financial amounts they would receive for their participation and the conditions that were indicated to them prior to their participation.

The sociodemographic questionnaire focused on basic aspects of gender, grade, age, major and their level of preference and exposure to Pipa music.
Next, I will discuss the instruments used, starting with the Musical Emotional Processing Scale, a self-report scale based on the two-dimensional model of valence and activation; showing good reliability in its application in research. Another instrument used, the Teacher Quality Assessment Scale for its application at the end of the class, some items were modified or eliminated, and finally 6 items were added (between 5 and 30) to constitute the evaluation range. A high value is also recorded in the reliability test, with a Cronbach alpha value of 0.91.
Paragraph 2.2.4. details basic aspects in:
• Selection and preparation of materials, through the use of 10 instrumental pieces of pure pipe, composed in tonic mode and interpreted by renowned artists. Selected fragments were classified by valence and energy. Included were both positive and negative fragments that would be associated with various emotions: positive more related to pleasure and commitment and negative associated with catharsis and creativity. In addition, a non-invasive neuroimaging technique was used as an instrument to measure changes in the level of hemoglobin oxygenation in the cerebral cortex (fNIRS - Functional Near-Infrared Spectroscopy).

  • The phases of the formal experiment were 3:
    1. Holistic listening.
    2. Teaching phase, according to the designated methods: master lesson and "built-in" method of Delacroze.
    3. Integral listening phase, completing at the end of the same a subjective assessment of the valence and emotional activation of each selected piece.
    I believe that the rewards given to participants, students and teachers (as indicated between lines 339 and 340) should be clearly identified in the document.
    I wish to highlight the appropriateness and relevance of the graphs and figures used, in particular, the figures describing the research processes (Figures 2b and 2c, in particular).
    Analysis of data.
    The bulky section on data analysis, supplemented by the accompanying archive, makes the analysis of this article not only a complex task but also an extensive one. There is no doubt that the statistics used have been appropriate for detecting the validity of comparisons between groups and teaching methods.
    In section 2.5.2. points, in addition, the comparison between different groups through ANOVA analysis and coherence wavelet transform (WTC), used to identify neurophysiological timing patterns that vary over time and to detect how signals relate based on their frequency and time scale, overcoming the limitations of correlation methods that assume static relationships.

Results.
Let us look at the most relevant aspects of the results recorded in this research, associating them with the hypotheses put forward.
Hypothesis1a. There is partial support for it, with the two experimental groups showing superior performance in the emotional valence of the experience. , although only the body movement group achieved significantly higher levels than the control group in both emotion recognition and experience. In a nutshell, they point out that emotion-centred instruction improves the processing of musical emotions more effectively than passive listening; that the body movement group exhibited a significantly higher emotional experience than the lecture group; Results consistent with cognitive theories of the body suggesting that physical engagement facilitates emotional involvement.
Hypothesis 1b.
Analysis of repeated measures ANOVA confirmed a significant effect of the teaching method focused on the body rhythm in line with the proposal put forward in this hypothesis.
Hypothesis 2a.
The recorded data can be considered partially confirmed The body rhythm group has shown, as postulated in the hypothesis a greater activation in all recorded channels. On the other hand, the master lesson group has not shown an increase in activation in the post-test which would confirm the cognitive theories that support greater activation when it corresponds to a clear body involvement.
Hypothesis 2b.
Scenario 2b is also partially supported. The data suggest that the teaching of the built-in rhythm strengthens the teacher-student neural coupling more effectively than didactic instruction, highlighting its role in facilitating emotionally resonant learning processes.

Discussion.

In this section, the authors continue with the process of comparison begun in the previous section. In summary, the higher emotional involvement and perception of teaching quality has been corroborated by the application of the body rhythm method, which has resulted in greater activation in several key brain regions associated with social emotions. Findings that highlight the connection of neural teacher-student coupling through the application of a teaching method based on body rhythm.

Implications and conclusions.

Based on the study data, mention should be made of the possibility of transferring these findings to wider educational fields than music education; the value of the research proposed for review in general pedagogy and the exploitation of emotions in any educational process.

Limitations and future work.

The authors are interested in broadening the range of body movements and musical parameters, refining the theoretical and practical framework of this method and improving its feasibility and applicability in real classroom environments as variables to be considered for further research.

Bibliography.

While acknowledging the quality of the bibliographical contributions, I believe that authors should include a link which allows readers easier access to their sources. On the other hand, some sources mentioned in the text do not appear in the final listing, which I consider an aspect to be corrected; in particular, I can indicate the mentions of Fuentes-Sánchez et al., 2020 and Nyclíček & Wing, 2024; as well as the position in the alphabetical listing of Palmiero et al., 2023.

Finally, I would like to highlight the high quality of the graphs and figures provided, as well as the complete list of statistics for presentation, both in the document for publication and in the attachment with additional data.

Author Response

  We are deeply grateful to the reviewer for their thorough, insightful, and constructive feedback on our manuscript. The positive comments on the theoretical foundation, experimental design, and graphical presentation are greatly encouraging. We have carefully considered all the points raised and have revised the manuscript accordingly to address each suggestion. Below, we provide a point-by-point response to the comments.

Comments 1: To propose more detail on the compensation received by students... clearly identify the rewards given to participants, students and teachers.

Response 1: We appreciate the reviewer's suggestion to enhance methodological transparency. Specific details have been incorporated into the revised manuscript (Section 2.4, Procedure). The revised wording is as follows: “All student participants received a compensation of 40 RMB upon completion of the entire experiment. The three instructors involved in teaching were also provided with a corresponding fixed honorarium. All details regarding compensation were clearly communicated to every participant as part of the informed consent procedure prior to the commencement of the study.”

Comments 2: Some sources mentioned in the text do not appear in the final listing... and the position in the alphabetical listing of Palmiero et al., 2023, and should include the source of the link.

Response 2: We sincerely thank the reviewer for their meticulous attention to the details of the reference list and for bringing these omissions and formatting issues to our attention. We sincerely apologize for these oversights in our initial submission. In response to this comment, we have taken the following comprehensive actions to ensure the accuracy and accessibility of the bibliography: (1) Added Missing References. The references for Fuentes-Sánchez et al., 2021 and Ny & Wing, 2024 have now been included in the revised reference list. (2) Corrected Alphabetical Order. The entire reference list has been carefully checked and reformatted to ensure strict alphabetical ordering by the first author's surname. This includes verifying the position of Palmiero et al., 2023. (3) Enhanced Accessibility. In accordance with the reviewer's suggestion to facilitate access for readers, we have now added stable DOIs as active, hyperlinked URLs for all applicable references in the list.

We believe these revisions have fully addressed the reviewer's concerns and have significantly improved the completeness and professionalism of the manuscript.

Comments 3 : The possibility of extending these findings beyond music education to broader educational fields should be mentioned... particularly in the realm of general pedagogy.

Response 3: We are grateful to the reviewer for insightful suggestion regarding the broader implications of our work. We fully agree that the findings on embodied learning (via the body rhythm method) and enhanced teacher-student neural coupling hold significant promise for general pedagogy beyond the domain of music education. In response, we have expanded the following content in Section 4.4 “Practical Implications” of the paper to clearly articulate the generalizability of our findings:

“ More importantly, this study reveals a potentially universal teaching mechanism: by mobilizing the fundamental social learning mechanism of imitation and synchronization, teaching methods can effectively activate students' neural and cognitive resources during the learning process, thereby deepening their emotional and cognitive engagement. This finding applies not only to musical emotion processing but also holds significant implications for the theoretical development and paradigm innovation of general pedagogy. Therefore, its applications extend far beyond the musical domain (e.g., instrumental/vocal instruction, music therapy) and can directly inform and optimize any discipline requiring high emotional investment and social interaction, such as language arts or physical education. This research not only addresses the urgent need to enhance students' emotional processing abilities in music education but also contributes a transferable core component for constructing an evidence-based, emotion-oriented interdisciplinary teaching framework.”

Once again, we extend our sincere gratitude to the reviewer for their valuable time and insightful comments, which have undoubtedly helped us to improve the quality and clarity of our manuscript.
